# LATENT SCORE-BASED REWEIGHTING FOR ROBUST CLASSIFICATION ON IMBALANCED TABULAR DATA

## ABSTRACT

Machine learning models often perform well on tabular data by optimizing average prediction accuracy. However, they may underperform on specific subsets due to inherent biases and spurious correlations in the training data, such as associations with non-causal features like demographic information. These biases lead to critical robustness issues as models may inherit or amplify them, resulting in poor performance where such misleading correlations do not hold. Existing mitigation methods have significant limitations: some require prior group labels, which are often unavailable, while others focus solely on the conditional distribution $P(Y|X)$, upweighting misclassified samples without effectively balancing the overall data distribution $P(X)$. To address these shortcomings, we propose a latent score-based reweighting framework. It leverages score-based models to capture the joint data distribution $P(X, Y)$ without relying on additional prior information. By estimating sample density through the similarity of score vectors with neighboring data points, our method identifies underrepresented regions and upweights samples accordingly. This approach directly tackles inherent data imbalances, enhancing robustness by ensuring a more uniform dataset representation. Experiments on various tabular datasets under distribution shifts demonstrate that our method effectively improves performance on imbalanced data.

## 1 INTRODUCTION

Machine learning applied to tabular data has achieved significant success across various practical domains (Liu et al., 2024). While models trained using empirical risk minimization (ERM) often perform well on average, achieving low test errors overall, they can still exhibit high error rates on specific subsets of data (Vapnik, 1999). This inconsistency highlights a critical robustness issue, primarily stemming from inherent biases in the data. For example, training data may contain spurious correlations where target labels are statistically linked to non-causal features like demographic information. As a result, models trained on such biased data may inherit or even amplify these biases, leading to poor performance on data subsets where these correlations do not apply.

To tackle this issue, researchers have proposed various methods to mitigate biases in the training data. Some approaches utilize additional prior information, such as group labels, to reduce the impact of spurious correlations by resampling or adding regularization terms (Arjovsky et al., 2019; Sagawa* et al., 2020). Unfortunately, in many practical situations, such prior information is incomplete or unavailable, limiting these methods' usefulness. To overcome this, other methods focus on automatically generating proxy information for debiasing (Nam et al., 2020; Liu et al., 2021a; Qiu et al., 2023). A common strategy involves pre-training a classification model on the training data and then upweighting samples that the model classifies incorrectly. Since these misclassifications are often determined by a sample's proximity to the model's decision boundary, we refer to these as boundary-based methods. While these methods can enhance robustness by focusing on underrepresented training samples near the classification boundary, their effectiveness is constrained as they may fail to achieve a globally balanced distribution of training data.

To illustrate the limitations of boundary-based methods, consider a binary classification problem with two features, $x_0$ and $x_1$, where the true classification boundary follows a sine curve (see Figure 1a). The training data is heavily concentrated in regions where $x_0 \in \left[\frac{\pi}{4}, \frac{3\pi}{4}\right] \cup \left[\frac{5\pi}{4}, \frac{7\pi}{4}\right]$, resulting in a highly imbalanced dataset, as visualized in Figure 1d. Ideally, reweighting should correct this

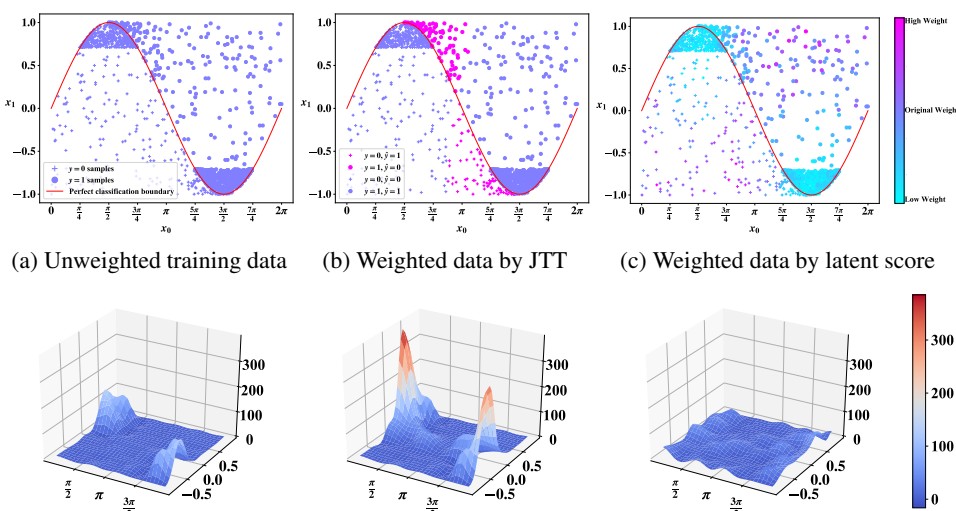

(a) Unweighted training data   (b) Weighted data by JTT   (c) Weighted data by latent score

(d) Unweighted data density (e) Weighted density with JTT (f) Weighted density with latent score
(classifier accuracy at 60.25%) (classifier accuracy at 69.50%) (classifier accuracy at 75.75%)

Figure 1: The comparison of weighted density plots on a synthetic dataset.

bias, yielding a more uniformly distributed dataset. Boundary-based methods like JTT (Liu et al., 2021a) try to achieve this by assigning higher weights to samples with large classification errors (Figure 1b). However, many reweighted samples are still from high-density regions near the decision boundary, rather than the low-density regions where the imbalance is most severe. The resulting distribution (Figure 1e) shows that the boundary-based method does not fully balance the training data. Consequently, the accuracy improvement on the balanced test set is modest, increasing from 60.25% to only 69.50%.

The fundamental limitation of boundary-based methods is their exclusive focus on the pre-trained classification boundary, $P(Y|X)$, neglecting the overall data distribution. This oversight leads to ineffective data balancing. Motivated by this observation, we aim to develop new approaches that not only consider classification errors but also more effectively address inherent data imbalances, ensuring a balanced and uniform distribution across the dataset.

To more effectively address inherent data imbalances, we propose leveraging score-based models, also known as diffusion models (Song & Ermon, 2019; Ho et al., 2020). These models can capture the joint data distribution $P(X, Y)$, providing a powerful tool for modeling the underlying data structure. Our key idea is to use a score-based model to estimate the density of samples in the dataset. By identifying low-density regions—areas where data is underrepresented—we can upweight samples from these regions, ensuring a more balanced and unbiased representation during training. This approach moves beyond boundary-based methods by directly targeting data imbalance, aiming to improve performance across diverse data subsets.

However, directly using density estimates from the score-based model poses challenges due to potential extreme values, leading to "density explosions" for certain samples. These extremes can overemphasize a few extremely high-density points, making it difficult to distinguish between other high- and low-density samples. To overcome this, we propose using a proxy for density based on an important observation. We demonstrate this observation using data from a mixture of two Gaussians, as shown in Figure 2. Regions with higher probability densities are represented by warmer colors in Figure 2a. To achieve a balanced data distribution, these regions are expected to have lower weights, as shown in Figure 2b. In Figure 2c, we illustrate that in high-density regions (e.g., points $B$ and $D$), the score vectors (pink arrows) of neighboring samples tend to align with the direction toward the high-density sample (e.g., cyan arrows $\overrightarrow{B_iB}$, where $i = 1, 2, ...$). In contrast, samples in low-density regions (e.g., points $A$ and $C$) do not exhibit this similarity. Essentially, if sample $B$ has higher density than sample $A$, the similarity between the score vectors and $\overrightarrow{B_iB}$ will generally be greater than that for $\overrightarrow{A_iA}$. This directional similarity serves as a proxy for density. By using this proxy, we avoid instability from extreme values, enabling a more stable and effective data reweighting approach.

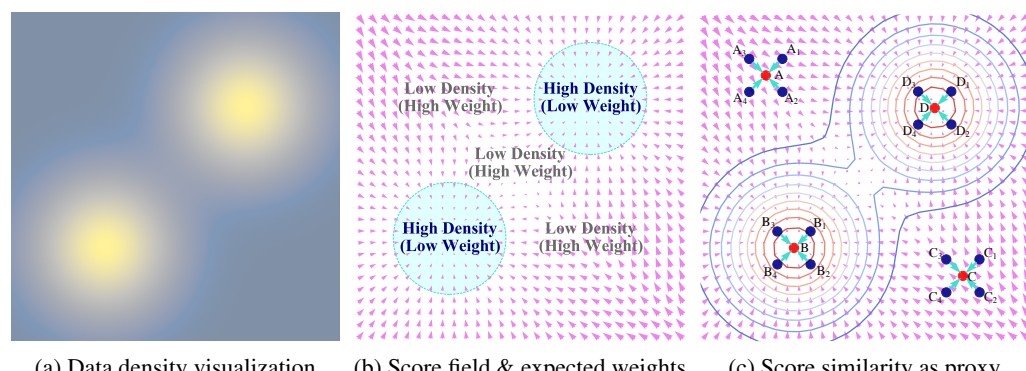

(a) Data density visualization     (b) Score field & expected weights     (c) Score similarity as proxy

Figure 2: The score fields of a mixture of two Gaussian.

Our method offers two key advantages. First, it requires no additional prior information, such as group labels, making it applicable in scenarios where such information is unavailable or incomplete. This flexibility allows broad adoption without relying on external data or assumptions. Second, our approach faithfully represents the joint data distribution $P(X, Y)$. By leveraging score-based models, we overcome the limitations of pre-trained classification boundaries $P(Y|X)$, which often inherit biases from the training process. This ensures that the final classification model remains unbiased and accurately captures the underlying relationships between features and labels, leading to improved performance and robustness.

In summary, our contributions are: (1) We recognize that improving robustness requires modeling the joint data distribution, a gap in current boundary-based methods that do not capture the true data distribution. (2) We introduce a new framework that accurately reflects the joint data distribution, detailed in Section 3. (3) We assess our method on various tabular datasets under distribution shifts, demonstrating through extensive experiments that our approach effectively enhances robustness.

## 2 RELATED WORKS

**Achieving robustness with prior information.** Commonly, researchers aim to train robust models towards distribution shift. Some works train models with the help of given or self-generated domain labels (Sagawa* et al., 2020; Arjovsky et al., 2019; Sun & Saenko, 2016; Liu et al., 2021b;c; Tong et al., 2023; Zhang et al., 2024b). Generally they expect the model's performance in the worst domain acceptable. Therefore, they may add regularizer into training loss when treating domain labels as extra supervised information. Predefining the form of prior data distribution is also a common approach (Shen et al., 2020; Duchi & Namkoong, 2021; Shen et al., 2023; Gu et al., 2024). Among them, stable learning is an effective approach which pursues feature independence under certain assumptions. The details are listed in Section A.9.

**Achieving robustness without prior information.** Apart from introducing the prior information, some other methods turn to use the performance of models to guide the final training process (Liu et al., 2021a; Nam et al., 2020; Levy et al., 2020; Qiu et al., 2023). They typically contain two stages. In the first stage, they identify some biased samples which will make ERM-based models make false predictions. In the second stage, they assign greater weights to these samples to develop a more robust model. However, we notice that few of the following methods model the joint data distribution, which is not fit for seeking overall robustness. The requirement of some prior-based methods is also rather strict and can not generalize to realistic scenarios.

**Generative Models for Robustness.** Generative models can produce novel and diverse data, enriching training datasets and leading to models with a deeper understanding of semantic content. Consequently, several studies have leveraged generative models to enhance robustness, primarily through data augmentation techniques (Li et al., 2021; Choi et al., 2021; Ilse et al., 2020; Dendorfer et al., 2021; Oberdiek et al., 2022; Zhang et al., 2022; 2024a). These approaches have demonstrated promising performance improvements. In contrast to these methods, we introduce a novel approach that utilizes generative models to estimate the density of the data distribution. By assigning weights to data samples based on this density estimation during training, we enhance the model's robustness.

## 3 METHOD

We present an unbiased learning framework that operates without prior information or pre-trained classification boundaries. The model is designed to provide robust predictions across a range of biased covariates. To address distributional shifts, we employ score-based methods to faithfully capture the joint distribution. The framework follows three steps: (1) training diffusion models on latent representations to model the data distribution, (2) sampling several timesteps and estimating scores to align the probability density of training data, and (3) reweighting samples based on data density to ensure a balanced distribution. The detailed procedure is provided in Algorithm 1.

### 3.1 PRELIMINARY

We first introduce the background on score models (Song & Ermon, 2019). Score models can model data distributions by learning score (*i.e.*, the gradients of probability density), and have shown remarkable performance in generative tasks. Song et al. proposed a unified framework based on Itô stochastic differential equations (SDEs). Training a score model typically involves an iterative forward and backward process. In the forward pass, a complex data distribution is gradually transformed into a Gaussian distribution by progressively adding noise, described by the following SDE:

$$\mathrm{d}\mathbf{x} = \mathbf{f}(\mathbf{x}, t)\mathrm{d}t + g(t)\mathrm{d}\mathbf{w}, \tag{1}$$

where $\mathbf{x} \in \mathbb{R}^d$ with $\mathbf{x}_0 \sim p_0$ representing the data distribution, $t \in [0, T]$, $\mathbf{f} : \mathbb{R}^d \times [0, T] \to \mathbb{R}^d$, $g : [0, T] \to \mathbb{R}$, and $\mathbf{w} \in \mathbb{R}^d$ is a standard Wiener process. The backward process reconstructs the original data structure from noisy data. Song et al. also introduced the corresponding "probability flow" ordinary differential equation (ODE):

$$\mathrm{d}\mathbf{x} = \left[\mathbf{f}(\mathbf{x}, t) - \frac{1}{2}g(t)^2 \nabla_{\mathbf{x}} \log p_t(\mathbf{x})\right] \mathrm{d}\bar{t}, \tag{2}$$

where $\bar{t}$ represents time flowing backward from $T$ to 0. We use a neural network, $s_\theta(\mathbf{x}_t, t)$, to estimate the score of the transformed data distribution at time $t$, $\nabla_{\mathbf{x}} \log p_t(\mathbf{x})$. The training loss for $s_\theta(\mathbf{x}_t, t)$ is defined through denoising score-matching:

$$\mathbb{E}_{t \sim \sigma(t)} \lambda(t) \mathbb{E}_{\mathbf{x}_0 \sim p_0} \mathbb{E}_{\mathbf{x}_t \sim p_{t|0}(\cdot|\mathbf{x}_0)} \left[\|s_\theta(\mathbf{x}_t, t) - \nabla_{\mathbf{x}_t} \log p_{t|0}(\mathbf{x}_t|\mathbf{x}_0)\|_2^2\right], \tag{3}$$

where $\sigma(t)$ represents the time variable distribution, and $\lambda(t)$ is a positive weighting function that stabilizes the time-dependent loss magnitude (Song et al., 2021). The diffusion process generally employs Gaussian transition kernels, leading to $p_{t|0}(\mathbf{x}_t|\mathbf{x}_0) = \mathcal{N}(\boldsymbol{\mu}_t, \sigma_t^2 \mathbf{I})$.

In summary, score models aim to compute score at each time scale with different noise, and finally reconstruct the clean sample with the guidance of score (Karras et al., 2022; Xu et al., 2022; 2023).

### 3.2 TRAINING DISTRIBUTION MODELING

To ensure robustness, we first train score models to approximate the probability distribution. To enable training on limited computational resources, we adopt the approach of latent diffusion (Rombach et al., 2022; Zhang et al., 2024c). Specifically, we first train a variational autoencoder (VAE), $\phi_{\mathcal{Z}} = \phi_{Enc} \cdot \phi_{Dec}$, using a $\beta$-VAE (Higgins et al., 2017), where the coefficient $\beta$ balances the reconstruction loss and the KL-divergence loss:

$$\mathcal{L}_{\phi_{\mathcal{Z}}} = \ell_{recon}(\mathbf{x}, \hat{\mathbf{x}}) + \beta \ell_{kl}. \tag{4}$$

We then obtain the latent representations using the trained encoder $\phi_{Enc}$. The subsequent diffusion process operates on these latent representations, $\mathbf{z} = \phi_{Enc}(\mathbf{x})$, rather than the raw data. Leveraging the VAE allows us to model the latent joint distribution of meaningful semantics, rather than superficial features in the raw data.

After the VAE model is trained, we use score-based methods to model the underlying distribution $p(\mathbf{z})$ (Song et al., 2021; Zhang et al., 2024c):

$$\mathbf{z}_t = \mathbf{z}_0 + \sigma(t)\boldsymbol{\varepsilon}, \ \boldsymbol{\varepsilon} \sim \mathcal{N}(\mathbf{0}, \mathbf{I}), \qquad \text{(Forward Process)} \tag{5}$$

$$\mathrm{d}\mathbf{z}_t = -2\dot{\sigma}(t)\sigma(t)\nabla_{\mathbf{z}_t} \log p(\mathbf{z}_t) \, \mathrm{d}t + \sqrt{2\dot{\sigma}(t)\sigma(t)} \, \mathrm{d}\boldsymbol{\omega}_t, \qquad \text{(Reverse Process)} \tag{6}$$

where $\mathbf{z}_0 = \mathbf{z}$ is the initial embedding from the encoder, $\mathbf{z}_t$ is the diffused embedding at time $t$, and $\sigma(t)$ is the noise level. In the reverse process (Eq. 6), $\nabla_{\mathbf{z}_t} \log p(\mathbf{z}_t)$ represents the score function of $\mathbf{z}_t$. Following Zhang et al., we set the noise scale $\sigma(t) = t$, making it linear with respect to time.

Following the approach of EDM (Karras et al., 2022), we train our neural network $F_\theta$ to directly predict the output at a given timestep $t$, rather than a scaled unit variance term $\sigma(t)\varepsilon$. The neural network is preconditioned with a $\sigma$-dependent skip connection, defined as:

$$D_\theta(z, \sigma) = c_{skip}(\sigma)z + c_{out}(\sigma)F_\theta(c_{in}(\sigma)z, c_{noise}(\sigma)), \tag{7}$$

where $c_{skip}(\sigma)$ modulates the skip connection between timesteps, $c_{in}(\sigma)$ and $c_{out}(\sigma)$ scale the input and output magnitudes, and $c_{noise}(\sigma)$ maps the noise level $\sigma$ into a conditioning input for $F_\theta$. The specifics of these scaling factors are detailed in Appendix A.2. The final training loss is:

$$\mathcal{L}(D_\theta, \sigma) = \mathbb{E}_t \lambda(t) \mathbb{E}_{z \sim p(\mathbf{z})} \|D_\theta(z + \sigma(t)\varepsilon, t) - z\|_2^2. \tag{8}$$

Here, $\lambda(t)$ is a positive weighting function to maintain the time-dependent loss at a consistent magnitude. The entire process uses the covariates $\mathbf{x}$ as input. After obtaining the latent representation $\mathbf{z}$, we partition $\mathbf{z}$ by their corresponding label $y$. For binary classification, two models are trained separately on $\mathbf{z}_{y=0}$ and $\mathbf{z}_{y=1}$ using Eq. 8. This allows us to model the joint training distribution for each class through these score models. We demonstrate the advantages of class-wise data separation and independent score model training in Section 3.3.

### 3.3 PROBABILITY DENSITY PROXY

At this stage, our goal is to estimate the probability density of each sample using the score models trained in Section 3.2. We first address the limitations of computing exact log-likelihood via the probability flow ordinary differential equation (ODE) proposed in (Song et al., 2021). Based on the insights from Figure 2, we introduce an approach to estimate the **relative** probability density through a similarity measure. The difference of score similarities acts as a proxy, preserving the relative magnitudes of probability densities among samples and offering a controllable numerical range at the model level. This makes it well-suited for computing weights, as described in Section 3.4.

#### 3.3.1 EXACT LOG-LIKELIHOOD COMPUTATION VIA PROBABILITY FLOW ODE

Since our goal is to distinguish data based on high or low probability density, a natural approach is to compute the exact log-likelihood using the trained score model from Section 3.2. Song et al. have proposed an estimation method for this. The forward diffusion process is represented by a stochastic differential equation (SDE) that gradually transforms a complex data distribution into a known prior distribution by injecting noise, as described in Eq. 1. The corresponding "probability flow" ordinary differential equation (ODE) is given in Eq. 2. By replacing the score $\nabla_{\mathbf{x}} \log p_t(\mathbf{x})$ with a neural network $\mathbf{s}_\theta(\mathbf{x}, t)$, the probability flow ODE takes the form:

$$d\mathbf{x} = \underbrace{\left[ \mathbf{f}(\mathbf{x}, t) - \frac{1}{2} g(t)^2 \mathbf{s}_\theta(\mathbf{x}, t) \right]}_{=: \tilde{\mathbf{f}}_\theta(\mathbf{x}, t)} d\bar{t}. \tag{9}$$

Using the instantaneous change of variables formula (Chen et al., 2018), the log-likelihood of $p_0(\mathbf{x})$ can be computed as:

$$\log p_0(\mathbf{x}(0)) = \log p_T(\mathbf{x}(T)) + \int_0^T \nabla \cdot \tilde{\mathbf{f}}_\theta(\mathbf{x}(t), t) \, dt, \tag{10}$$

where the random variable $\mathbf{x}(t)$ as a function of $t$ is obtained by solving the ODE in Eq. 9.

However, this estimation method has a limitation: extreme differences in log-likelihood values lead to highly imbalanced sample weights, failing to reflect the **relative magnitudes** of densities among different samples globally. Because the log-likelihood strictly follows the functional form of the original data distribution, it leads to explicit numerical differences. When only a few data points have significantly high probability densities, their log-likelihoods become much higher than those of other samples. Using these likelihoods to compute sample weights causes high-density samples to disproportionately overshadow others, whether those have low or moderately high densities. This

imbalance diminishes distinctions among samples outside the highest-density regions, reducing the ultimate training on the remaining samples to an unweighted process without clear density differentiation. To validate this, we visualize the probability density using log-likelihood on a synthetic dataset in Section 4.6. In summary, although log-likelihood faithfully reflects the original density, it does not meet the requirement for global relative density comparison needed for sample reweighting.

### 3.3.2 DATA DENSITY ESTIMATION VIA SIMILARITY DIFFERENCE MEASURE

Because the exact log-likelihood is unsuitable for sample reweighting, we adopt an alternative measure for global density modeling. As discussed in Section 1, we estimate the relative probability density by measuring similarities between the scores of noisy points. According to the forward process in Eq. 5, a clean point $z_0$ transforms into a noisy point $z_t$ by adding noise scaled by $\sigma(t)$. This mirrors the scenario depicted in Figure 2, where point $A$ moves to its noisy neighbor $A_i'$. The vector $\overrightarrow{AA_i'}$ corresponds to $\sigma(t)\varepsilon$, and its length $|\overrightarrow{AA_i'}|$ represents the magnitude of the added noise $\|\sigma(t)\varepsilon\|_2$. To sample a neighborhood of noisy points around a data point $z_i$, we select $T$ fixed timesteps $t_0, t_1, \ldots, t_{T-1}$ and add noise as per Eq. 5. Repeating this process $K$ times yields $T \times K$ noisy points. We then estimate the relative probability density using these points, employing the squared error as the similarity metric. For a sample $z_i$ and a preconditioned network $F_\theta$, we compute the aggregated similarity, *i.e.*, the average similarity across several noise scales, as:

$$\text{Sim}(z_i; F_\theta) = \frac{1}{K} \sum_{k=1}^{K} \mathbb{E}_t \left[ \lambda(t) \, \| D_\theta(z_i + \sigma(t)\varepsilon, t) - z_i \|_2^2 \right]. \tag{11}$$

The key difference between Eq. 8 and Eq. 11 is that $\text{Sim}(z_i; F_\theta)$ samples only from specific fixed and sparse timesteps, accounting for different noise scales. Moreover, we randomly sample these timesteps multiple times and average the results to obtain a robust similarity measure.

Using Eq. 11, we compute the aggregated similarity across all training data points to indicate their densities. However, in addition to shifts in latent covariates $x$, $y$-shift also commonly occurs in real-world data. To address this, we model the data distribution for each class separately (see Section 3.2) and use the difference in aggregated similarities as a proxy for the final relative probability density. For example, for a data point $z_i$ with class label $y_{z_i}$, we compute $\text{SimDiff}(z_i)$ as:

$$\text{SimDiff}(z_i) = \text{Sim}(z_i; F_{y \neq y_{z_i}}) - \text{Sim}(z_i; F_{y = y_{z_i}}). \tag{12}$$

$\text{Sim}(\cdot)$ allows us to differentiate the densities of samples within a specific class, where a larger $\text{Sim}(\cdot)$ indicates a lower density. The $\text{SimDiff}(\cdot)$ measure further accounts for distribution shifts in the label $y$ while preserving the properties of $\text{Sim}(\cdot)$ that reflect $p(z)$. When the number of training samples for a given class $y_k$ is small, $\text{Sim}(\cdot; F_{y=y_k})$ increases because the score model $F_{y=y_k}$ is trained on a narrower and less comprehensive data space. Consequently, $F_{y=y_k}$ finds it more challenging to guide noisy points back to their original locations via score fields, leading to higher estimation errors in $\text{Sim}(\cdot; F_{y=y_k})$. The subtraction operation thus causes samples from the minority class to have a lower $\text{SimDiff}(\cdot)$ compared to those from the majority class. In this way, SimDiff captures both covariate distribution shifts in $p(x)$ ($p(z)$) via $\text{Sim}(\cdot)$ and label distribution shifts in $p(y)$ through the subtraction operation, providing a means to indicate relative probability density.

In summary, a lower $\text{SimDiff}(\cdot)$ indicates a lower probability density. Whether the imbalance arises from covariates $x$ or labels $y$, $\text{SimDiff}(\cdot)$ consistently and faithfully reflects the relative probability density. The whole computation process requires no prior information or predefined boundaries.

### 3.4 UNBIASED LEARNING ON DISTRIBUTION-BALANCED DATA

Our ultimate goal is achieving the robustness for all sensitive covariates $x$. The implicit joint distribution on latent representation $z$ has been modeled in Section 3.2 and Section 3.3 through score models. Therefore, we could conduct sample reweighting guided by Eq. 12 directly to acquire an overall unbiased training distribution. Each $z_i$ is assigned a weight $w_i$ through

$$w_i = \frac{\exp(-\text{SimDiff}(z_i)/\tau)}{\sum_{j=0}^{N-1} \exp(-\text{SimDiff}(z_j)/\tau)}, \tag{13}$$

where $N$ is the number of all training samples and $\tau$ denotes a temperature which controls the scale of reweighting. Finally, we train an unbiased classification model $\psi$ as:

$$\mathcal{L}_{\text{classification}} = \mathbb{E}_{(z_i, y_i)}[w_i \ell(\psi(z_i), y_i)], \tag{14}$$

where $\ell$ stands for cross entropy loss. When testing, we only use $\phi_{Enc}$ and $\psi$ to make predictions.

## 4 EXPERIMENT

To evaluate the effectiveness of our method, we conduct extensive experiments in various settings.

### 4.1 DATASETS

To comprehensively validate our method, we selected six diverse datasets that exhibit various types of distribution shifts, including covariate shifts and concept shifts.

The details of our used datasets are as follows:

- **Adult** (Becker & Kohavi, 1996): The classification goal is to predict whether income exceeds $50K/yr based on census data.
- **Bank** (Moro et al., 2012): The data is related with direct marketing campaigns of a Portuguese banking institution. The task is to predict if the client will subscribe a term deposit.
- **Default** (Yeh, 2016): This dataset aims at the case of customers' default payments in Taiwan. The task is to predict whether the client will default payment next month.
- **Shoppers** (Sakar & Kastro, 2018): The task is to predict if the user's session ends with the shopping behavior. The distribution shift mainly demonstrates as $y$-shift: About 84.5% samples were class samples that did not end with shopping, and the rest were positive class samples.
- **Taxi** (Navas, 2018): This dataset collects some information about the pickup and dropoff of taxi rides. The task is to predict whether the total ride duration time exceeds 30 minutes. We choose the data collected in Mexican City.
- **US-Wide ACS PUMS Data** (Ding et al., 2021): This large dataset contains individual records from US Census sources. We choose the task where the outcome is whether an individual's income exceeds 50k. The performance is validated separately on three randomly chosen states.

Specifically, for the Taxi and ACS datasets, we followed the preprocessing guidelines outlined in the `WhyShift` benchmark (Liu et al., 2024).

### 4.2 EVALUATION METRICS

Our objective is to ensure that the model consistently makes robust predictions across all non-causal covariates, rather than focusing on a single one. For each dataset, we select several deterministic non-causal attributes based on prior knowledge—for example, sex and race in the ACS income dataset. We then record the worst-case prediction results for each feature and compute the average of the worst-group accuracies across these features. Details of the selected features are provided in Appendix A.5. Additionally, we track the mean accuracy for these features. We aim for our model to enhance the average worst-group performance without significantly compromising mean accuracy.

### 4.3 IMPLEMENTATION DETAILS

We conducted each experiment three times with different random seeds and report the mean results in Table 1 and Table 2. The corresponding standard deviations are provided in Appendix A.3. Our prediction model consists of two components: a Variational AutoEncoder (VAE) for generating latent representations, and a MultiLayer Perceptron (MLP) for classification. Both our method and the baseline models utilize the same architecture. For training the VAE, we followed the default settings described in TabSyn (Zhang et al., 2024c).

In addition, our method involves training score models. Following the guidance from TabSyn, we use a 4-layer MLP architecture as the backbone for our score models. We also adopt the training paradigm from EDM (Karras et al., 2022), which mitigates the influence of varying noise scales on neural network training. The noise scale is set to increase linearly with time, *i.e.*, $\sigma(t) = t$, where $t$ follows a log-normal distribution, resulting in $\ln(\sigma(t)) \sim \mathcal{N}(P_{\text{mean}}, P_{\text{std}}^2)$. The $P_{\text{mean}}$ and $P_{\text{std}}$ are set to $-1.2$ and $1.2$, respectively. When computing aggregated similarity in Eq. 11, we choose 0.002 and 80 as the minimum and maximum values of $t$ and select $T$ timesteps with equal intervals between them. Regarding hyperparameter selection, we set $T$ to 10 and $\tau$ to 3 for all experiments.

| Methods | Adult | | Bank | | Default | | Shoppers | | Taxi | | Average | |
|---|---|---|---|---|---|---|---|---|---|---|---|---|
| | Mean | Worst | Mean | Worst | Mean | Worst | Mean | Worst | Mean | Worst | Mean | Worst |
| ERM | 71.30 | 48.18 | 70.23 | 40.13 | 62.98 | 36.12 | 77.12 | 53.91 | 67.53 | 59.48 | 69.83 | 47.56 |
| CVaR-DRO | 71.95 | 49.02 | 68.93 | 38.71 | 62.43 | 34.60 | 76.65 | 51.26 | 65.55 | 54.51 | 69.10 | 45.62 |
| $\chi^2$-DRO | 71.85 | 50.09 | 70.88 | 40.04 | 62.03 | 34.16 | 76.95 | 52.40 | 68.27 | 62.43 | 70.00 | 47.83 |
| KL-DRO | **74.33** | 49.17 | 69.95 | 39.89 | 59.03 | 19.39 | 79.53 | 59.74 | 62.30 | 47.93 | 69.03 | 43.22 |
| EIIL | 69.37 | 38.97 | 61.85 | 21.69 | **65.05** | 28.91 | 74.82 | 46.18 | **69.52** | 58.00 | 68.12 | 38.75 |
| JTT | 71.46 | 49.93 | 68.78 | 37.77 | 62.47 | 35.51 | 78.10 | 52.59 | 67.47 | 60.01 | 69.65 | 47.16 |
| FAM | 72.85 | 49.59 | **71.03** | 41.09 | 62.50 | 37.12 | 76.60 | 53.51 | 67.90 | 62.24 | 70.18 | 48.71 |
| SRDO | 71.27 | 46.44 | 66.34 | 32.08 | 62.38 | 33.34 | 76.24 | 53.11 | 64.57 | 58.18 | 68.16 | 44.63 |
| Ours | **74.33** | **54.79** | 69.50 | **41.28** | 62.62 | **38.78** | **79.68** | **60.73** | 67.85 | **63.14** | **70.80** | **51.74** |

Table 1: The classification results on five datasets which exhibit various kinds of distribution shifts. The **bold** and underline denote the best and the second best results respectively.

## 4.4 COMPARED BASELINES

As referred in Section 4.2, our goal is to enhance overall robustness, with the accuracy on the worst group serving as our main evaluation metric. Therefore, we compare our method with several robust machine learning techniques, including **ERM** (Vapnik, 1999), **CVaR-DRO** (Levy et al., 2020), $\chi^2$**-DRO** (Levy et al., 2020), **KL-DRO** (Duchi & Namkoong, 2021), **JTT** (Liu et al., 2021a), **EIIL** (Creager et al., 2021), **FAM** (Petzka et al., 2021; Zou et al., 2024) and **SRDO** (Shen et al., 2020). Among these, CVaR-DRO, $\chi^2$-DRO, KL-DRO, JTT, and SRDO employ reweighting processes similar to ours, while EIIL and FAM enhance robustness by adding regularization terms.

## 4.5 CLASSIFICATION RESULTS

### 4.5.1 ROBUSTNESS ACROSS VARIOUS DISTRIBUTION SHIFTS

We first demonstrate the effectiveness of our method under various types of distribution shifts in Table 1. Our method consistently achieves the highest worst-case accuracy across all five datasets, with an average improvement of at least 3% in worst-case accuracy over all baseline methods. Additionally, it attains the best mean accuracy on two of the five datasets, highlighting its ability to enhance robustness without a significant reduction in overall prediction accuracy.

In contrast, DRO-based methods like KL-DRO perform inconsistently on our datasets. For instance, while KL-DRO achieves the second-best accuracy on the Shoppers dataset, its performance deteriorates significantly on Default and Taxi. This inconsistency stems from KL-DRO's reliance on expanding the search space without adequately modeling the underlying data distribution.

Another popular invariant learning-based method, EIIL (Creager et al., 2021), achieves high mean accuracy. However, its worst-case accuracy is relatively disappointing compared to other methods. EIIL divides samples into different groups based on generated environment labels, which creates implicit boundaries based on data's inherent characteristics. While this boundary-based approach may improve overall accuracy by exploiting data biases, it undermines the balance and robustness required to handle sensitive non-causal attributes.

Compared to the above methods which rely on either prior information or boundaries, our method could achieve consistent and stable robustness on these datasets. The results demonstrate the benefit of modeling relative probability density as referred in Section 1.

### 4.5.2 GENERALIZATION CAPABILITY UNDER SELECTION BIAS

We further validate the generalization capability of our method with the ACS Income dataset. The key objective is to assess whether our model can learn a robust predictive function from a single source environment that generalizes well to others. We randomly select three geographically distant states in the U.S. and train classification models separately on data from each state.

The second to ninth columns of Table 2 show the results when the model is tested on data from its source state. We then evaluate each model on data from another state in a round-robin manner, with the results recorded in the tenth to last column. The observations are as follows:

| Methods | AZ | | MA | | MI | | Average | | AZ→MA | | MA→MI | | MI→AZ | | Average | |
|---|---|---|---|---|---|---|---|---|---|---|---|---|---|---|---|---|
| | Mean | Worst | Mean | Worst | Mean | Worst | Mean | Worst | Mean | Worst | Mean | Worst | Mean | Worst | Mean | Worst |
| ERM | 76.95 | 64.07 | 78.18 | **74.05** | 73.70 | 62.42 | 75.83 | 66.04 | 73.05 | 57.82 | 74.28 | 67.78 | 72.38 | 56.77 | 73.23 | 60.79 |
| CVaR-DRO | **77.00** | 64.41 | 77.85 | 72.85 | 73.48 | 61.88 | 76.11 | 66.39 | 74.10 | 59.42 | 74.85 | 68.57 | 71.75 | 54.67 | 73.57 | 60.88 |
| $\chi^2$-DRO | 76.95 | 64.07 | 77.55 | 71.82 | 73.13 | 59.03 | 75.88 | 64.97 | 74.53 | 61.18 | 74.23 | 68.53 | 71.33 | 53.05 | 73.36 | 60.92 |
| KL-DRO | 75.68 | 60.98 | 78.23 | 71.22 | **76.10** | 62.88 | 76.67 | 65.03 | 74.20 | 57.78 | 75.35 | 68.72 | 74.73 | 57.20 | 74.76 | 61.23 |
| EIIL | 74.98 | 55.22 | 78.18 | 68.42 | 75.60 | 63.77 | 76.25 | 62.47 | **75.35** | 56.80 | **76.92** | 65.62 | **74.78** | 57.32 | **75.68** | 59.91 |
| JTT | 75.70 | 57.73 | 77.48 | 68.48 | 74.48 | 60.63 | 75.88 | 62.28 | 73.90 | 55.17 | 74.00 | 66.57 | 72.65 | 55.17 | 73.52 | 58.97 |
| FAM | 75.00 | 57.82 | **78.73** | 70.12 | 74.38 | 59.45 | 76.03 | 62.46 | 72.93 | 54.60 | 74.68 | 66.35 | 73.40 | 54.53 | 73.67 | 58.49 |
| SRDO | 75.17 | 60.03 | 77.85 | 69.52 | 73.30 | 54.63 | 75.44 | 61.39 | 74.00 | 58.37 | 74.78 | 61.77 | 73.83 | 60.05 | 74.39 | 60.07 |
| Ours | 76.28 | **66.42** | 78.35 | 73.33 | 75.53 | **67.10** | **76.72** | **68.95** | 75.00 | 62.85 | 74.75 | 68.75 | 74.73 | 64.10 | 74.83 | **65.23** |

Table 2: The performance on ACS Income task. The **bold** and underline denote the best and the second best results respectively.

- For the single-source data experiments (Columns 2 to 10), our method achieves both higher mean accuracy and worst-case accuracy compared to most baselines. This consistency with the results in Table 1 supports the robustness of our approach.

- For the generalization experiment on different-source data (Columns 11 to 19), our method does not achieve the highest mean accuracy. This is due to the change in causal mechanisms across states (Liu et al., 2024), leading to variations in the predictive mechanism. However, our method still achieves strong worst-case accuracy, thanks to our modeled relative probability density. Our training process ensures fair treatment of covariates through score-based similarity, which is more reliable than the distances or boundaries employed by other methods.

## 4.6 VISUALIZATION OF SCORE-SIMILARITY-BASED WEIGHTS

In this section, we provide the visualization results on a synthetic dataset to directly explain the reweighting process of our methods. In addition, we will explain why we use our aggregated similarity difference measure in Eq. 12 instead of running a sampler based on the probability flow ordinary differential equation that allow for exact likelihood computation proposed by Song et al..

**Synthetic Data Generation.** Our imbalanced synthetic training data has been illustrated in Figure 1a. The data have two features, $x_0$ and $x_1$. The perfect classification boundary is a sine curve. We introduce the bias on training data by sampling more points in regions where $x_0 \in [\frac{\pi}{4}, \frac{3\pi}{4}] \cup [\frac{5\pi}{4}, \frac{7\pi}{4}]$. We expect the reweighting process could make the whole data distribution fairly balanced.

**Exact Likelihood Computation by Probability Flow ODE.** As referred in Section 3.3, we can compute the exact log-likelihood through a probability flow ODE in Eq. 10. Following Song et al., we apply the Skilling-Hutchinson trace estimator (Skilling, 1989; Hutchinson, 1990) to estimate $\nabla \cdot \tilde{\mathbf{f}}_\theta(\mathbf{x}(t), t)$ and finally achieves the log-likelihood.

**Extreme Value Problem Brought by Exact Likelihood.** The log-likelihood of $p(\mathbf{x})$ represents the estimated probability density at the feature level, making it useful for guiding the reweighting process. We compute the mean of the estimated probabilities across all features to determine the final sample-level probability density. In Figure 3a, color depth represents the log-likelihood of the predicted probability density. Points with higher density are shaded closer to violet, while those with lower density appear more blue. Interestingly, not all points in high-probability regions are assigned warm colors. In fact, most points are blue, similar to those in low-density regions. Only a few points located at cluster centers exhibit significantly higher log-likelihoods. Reweighting based on these estimates could lead to disproportionate emphasis on a small number of extreme high-density points, potentially obscuring the distinction between other high- and low-probability regions.

**Density-Aware Weights Computed from Our Similarity-based Measure.** We then visualize the unnormalized score similarity in Figure 3b. Points with lower error, corresponding to higher relative probability density, are shaded in blue and are expected to be downweighted. The visualization clearly distinguishes these points, with two regions of relatively high probability density appearing in blue. In contrast, points with truly low probability are represented by higher aggregated error, shown in purple, and can be easily upweighted. Our similarity measure not only differentiates density clearly but also maintains a consistent and stable numerical range, making it suitable for sample reweighting.

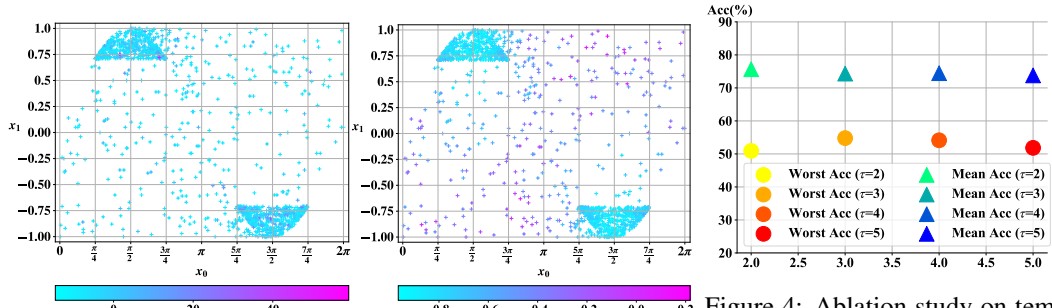

(a) Computed log likelihood of $p(\mathbf{x})$    (b) Score similarity as proxy

Figure 3: Our score similarity is a good proxy for density.

Figure 4: Ablation study on temperature of reweighting scale $\tau$. Our method is insensitive to $\tau$.

|  | $T = 5$ | | $T = 10$ | | $T = 15$ | | $T = 20$ | |
|---|---|---|---|---|---|---|---|---|
|  | Mean | Worst | Mean | Worst | Mean | Worst | Mean | Worst |
| Acc. (%) | $73.27 _{\pm 0.84}$ | $50.82 _{\pm 3.17}$ | $74.33 _{\pm 0.28}$ | $54.79 _{\pm 2.01}$ | $74.92 _{\pm 0.52}$ | $56.90 _{\pm 1.51}$ | $74.91 _{\pm 0.52}$ | $57.46 _{\pm 2.48}$ |

Table 3: Ablation studies of the number of chosen timesteps $T$ on Adult dataset.

In summary, compared to the exact log-likelihood calculation, our method more effectively captures relative probability densities across the dataset. By computing similarity, any two sample points with significantly different probability densities are clearly distinguished and distinctly colored. In contrast, the exact log-likelihood tends to overemphasize a few high-density points, obscuring the distinction between other high- and low-density samples.

## 4.7 ABLATION STUDIES

We conduct the sensitivity analysis in this section. As referred in Algorithm 1, our method has three hyperparameters. $T$ denotes the number of selected timesteps for computing aggregation similarity. $\tau$ controls the strength of reweighting based on SimDiff($\cdot$). $K$ represents the number of repeated sampling iterations for computing scores. The discussion on $K$ is deferred to Appendix A.4, as it does not functionally impact the score computation.

The ablation study results of $T$ on Adult are shown in Table 3. There are two main observations:

- The mean accuracy shows minimal variation with changes in $T$.
- The worst-case accuracy improves as $T$ increases. This is because $T$ indirectly represents the number of aggregated noisy points in the sampled neighborhood. A larger $T$ leads to a more accurate computation of aggregated similarity in Eq. 11. We set the default value of $T$ to 10, balancing fast computation with optimal performance.

In Figure 4, we show the classification results by using different $\tau$. Our model performs consistently well with all the values. The worst accuracies under different $\tau$ all surpass the baseline methods in Table 1, which proves that our method does not heavily rely on the values of hyperparameters.

## 5 CONCLUSION

In this paper, we tackle the challenge of improving model robustness, measured across all non-causal covariates rather than focusing on a single attribute. We observe that previous methods are limited by their inability to model the original joint data distribution and apply effective balancing strategies. To overcome these limitations, we propose using score-based models to capture the latent data distribution. Specifically, we estimate scores at several fixed timesteps and use their similarity to model the relative probability density of each sample. A score-based reweighting strategy is then employed to train a robust classification model. Our approach requires no prior information during training and ensures that the reweighting process aligns with the original data distribution. Experiments on seven datasets demonstrate that our method effectively balances the original training data globally and achieves robust performance under distribution shifts.

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

## A APPENDIX

The supplementary materials are structured as follows:

### A.1 THE PSEUDOCODE OF OUR METHOD

---

**Algorithm 1** The pseudocode of our method in binary classification problem.

---

**Input**: Training dataset $\mathcal{D} = \{x, y\} = \{x_i, y_i\}_{i=0}^{N-1}$, $T$ fixed noises along with specific timesteps $\{\sigma(t)\}_T$, the temperature of the reweighting scale $\tau$, the repeated sampling times $K$

**Parameters to be optimized**: a VAE $\phi_{\mathcal{Z}}(\cdot)$, two score-based models $\{s_j(\cdot, \cdot)\}, j \in \{0, 1\}$, a final classification model $\psi(\cdot)$

    **// Stage One: Training Distribution Modeling**
    Train a VAE $\phi_{\mathcal{Z}} = \phi_{Enc} \cdot \phi_{Dec}$ with Eq. 4
    Obtain latent representation with trained encoder: $z = \phi_{Enc}(x)$
    Separate $z$ into $z_{y=0}$ and $z_{y=1}$ according to their labels, train score model $s_j$ with $z_j$ separately by Eq. 8
    **// Stage Two: Probability Density Aligning**
    For each latent $z_i \in z$, compute the aggregated similarity with target class model $\text{Sim}(z_i; s_{y=y_{z_i}})$ and with non-target class model $\text{Sim}(z_i; s_{y \neq y_{z_i}})$ through Eq. 11
    Compute $\text{SimDiff}(z_i)$ through Eq. 12 to indicate $z_i$'s relative probability density
    **// Stage Three: Unbiased Learning on Distribution-balanced Data**
    Reflect SimDiff into weights $w_i$ through Eq. 13
    Train the final unbiased classification model $\psi$ with Eq. 14
    Return $\phi_{Enc}$ and $\psi$ for testing

---

### A.2 THE CHOICE OF NETWORK PRECONDITIONING

In our method, we follow the practice from EDM (Karras et al., 2022) to train our neural network. The key lies in using preconditioning techniques to make the output of neural network stable, instead of varying with the scale of variance $\sigma(t)\varepsilon$. We take the default choice of scaling factors from Karras et al.; Zhang et al.. The details are listed in Table 4[1].

### A.3 DETAILS OF THE STANDARD DEVIATION OF OUR EXPERIMENTS

We randomly conducted each experiment three times using different seeds and computed the mean classification results, as shown in Tables 1 and 2. The standard deviation across these three runs is provided in Tables 5 and 6, respectively.

---

[1]We set $\sigma_{\text{data}} = 0.5$ in our experiment as TabSyn (Zhang et al., 2024c) did.

| Skip scaling $c_{\text{skip}}(\sigma)$ | $\sigma_{\text{data}}^2/(\sigma^2 + \sigma_{\text{data}}^2)$ |
|---|---|
| Output scaling $c_{\text{out}}(\sigma)$ | $\sigma \cdot \sigma_{\text{data}}/\sqrt{\sigma_{\text{data}}^2 + \sigma^2}$ |
| Input scaling $c_{\text{in}}(\sigma)$ | $1/\sqrt{\sigma^2 + \sigma_{\text{data}}^2}$ |
| Noise cond. $c_{\text{noise}}(\sigma)$ | $\frac{1}{4}\ln(\sigma)$ |

Table 4: The choices of various scaling factors for denoiser $D_\theta(\cdot, \cdot)$ in Eq. 7.

| Methods | Adult | | Bank | | Default | | Shoppers | | Taxi | |
|---|---|---|---|---|---|---|---|---|---|---|
| | Mean | Worst | Mean | Worst | Mean | Worst | Mean | Worst | Mean | Worst |
| ERM | 0.47 | 1.64 | 0.35 | 1.85 | 0.21 | 3.93 | 0.73 | 2.46 | 0.14 | 2.61 |
| CVaR-DRO | 0.68 | 2.63 | 1.52 | 3.59 | 0.33 | 2.44 | 0.31 | 2.01 | 2.00 | 5.93 |
| $\chi^2$-DRO | 0.73 | 1.78 | 0.81 | 3.56 | 0.19 | 2.12 | 1.72 | 3.60 | 0.99 | 1.08 |
| KL-DRO | 0.14 | 0.90 | 0.60 | 1.75 | 3.25 | 5.10 | 1.08 | 3.98 | 8.96 | 23.68 |
| EIIL | 2.50 | 8.15 | 2.49 | 6.91 | 1.06 | 7.55 | 7.04 | 15.02 | 0.07 | 4.43 |
| JTT | 1.89 | 3.65 | 0.67 | 1.50 | 0.90 | 1.58 | 0.75 | 3.35 | 0.42 | 2.60 |
| FAM | 1.65 | 1.95 | 3.89 | 5.25 | 0.19 | 2.07 | 0.09 | 2.05 | 0.75 | 0.63 |
| SRDO | 2.43 | 6.82 | 2.55 | 5.44 | 8.06 | 10.94 | 0.35 | 2.31 | 2.07 | 2.50 |
| Ours | 0.28 | 2.01 | 0.39 | 1.56 | 0.12 | 1.22 | 1.44 | 3.39 | 0.16 | 2.02 |

Table 5: The standard deviation of the classification results across three runs on five datasets.

## A.4 THE ABLATION STUDIES OF REPEATED SAMPLING TIMES

In Section 4.7, we conduct sensitivity analysis on the number of selected timesteps $T$ and the temperature $\tau$, which controls the reweighting scale. These two hyperparameters directly influence score computation. Additionally, we have another hyperparameter, $K$, which governs the number of repeated sampling iterations. While $K$ does not functionally affect score computation, it impacts the robustness of our computed values. Therefore, we will analyze the effect of $K$ independently in this section to distinguish it from $T$ and $\tau$.

The experimental results are listed in Table 7. We could find larger $K$ corresponds to a longer computation time and a more stable performance. We choose 32 as the default value for $K$, which achieves a good trade-off between sampling time and model performance.

## A.5 DETAILS OF THE SELECTED NON-CAUSAL ATTRIBUTES FOR MEASURING OVERALL ROBUSTNESS

As referred in Section 4.2, we measure overall robustness by selecting several non-causal attributes and recording their worst-case prediction results. Here we provide the details of selected attributes.

- **Adult** (Becker & Kohavi, 1996): We select marital status, race, and sex as sensitive attributes.
- **Bank** (Moro et al., 2012): We select age, housing status, marital status, and the last contact duration as sensitive attributes.
- **Default** (Yeh, 2016): We select age, sex, and the amount of the given credit as sensitive attributes.
- **Shoppers** (Sakar & Kastro, 2018): We select the traffic type, the visitor type as returning or new visitor, a Boolean feature indicating whether the date of the visit is weekend as sensitive attributes.
- **Taxi** (Navas, 2018): We select the indicator for weekday, the month of picking up, and the direction as sensitive attributes.
- **US-Wide ACS PUMS Data** (Ding et al., 2021): We select race and sex as sensitive attributes.

We demonstrate the Pearson correlation coefficients between the attributes and target vairable in training and test data in Table 8.

We select sensitive attributes for evaluation based on the following criteria:

| Methods | AZ | | MA | | MI | | AZ → MA | | MA → MI | | MI → AZ | |
|---|---|---|---|---|---|---|---|---|---|---|---|---|
| | Mean | Worst | Mean | Worst | Mean | Worst | Mean | Worst | Mean | Worst | Mean | Worst |
| ERM | 0.85 | 1.31 | 0.46 | 1.23 | 0.49 | 2.66 | 0.78 | 1.62 | 0.32 | 2.50 | 0.67 | 1.82 |
| CVaR-DRO | 0.14 | 3.70 | 1.27 | 1.88 | 0.74 | 4.97 | 0.35 | 2.51 | 0.14 | 0.65 | 0.28 | 2.55 |
| $\chi^2$-DRO | 0.14 | 4.60 | 0.92 | 1.58 | 1.38 | 2.06 | 0.25 | 1.15 | 0.18 | 0.67 | 0.60 | 1.76 |
| KL-DRO | 1.23 | 3.00 | 0.32 | 2.41 | 0.07 | 0.37 | 1.20 | 2.91 | 0.78 | 1.71 | 0.25 | 0.97 |
| EIIL | 1.80 | 6.65 | 0.67 | 5.86 | 0.49 | 3.19 | 1.70 | 5.52 | 1.59 | 5.88 | 1.51 | 4.95 |
| JTT | 1.06 | 2.49 | 0.25 | 2.70 | 0.53 | 1.84 | 0.78 | 3.14 | 0.28 | 2.33 | 0.35 | 0.62 |
| FAM | 0.57 | 2.17 | 0.03 | 3.33 | 2.09 | 6.29 | 0.53 | 2.45 | 0.11 | 2.44 | 1.27 | 5.24 |
| SRDO | 2.22 | 10.23 | 1.01 | 2.49 | 1.22 | 5.75 | 1.83 | 8.57 | 0.49 | 4.26 | 0.35 | 0.96 |
| Ours | 0.11 | 0.89 | 0.14 | 1.48 | 1.77 | 1.61 | 0.07 | 0.40 | 0.35 | 0.30 | 0.35 | 0.35 |

Table 6: The standard deviation of the classification results across three runs on ACS dataset.

| $K$ | Mean (%) | Worst (%) | Running Time (s) |
|---|---|---|---|
| 8 | $73.97_{\pm 0.50}$ | $52.47_{\pm 0.71}$ | $7.35_{\pm 0.01}$ |
| 16 | $74.13_{\pm 0.67}$ | $54.18_{\pm 1.90}$ | $14.28_{\pm 1.50}$ |
| 32 | $74.33_{\pm 0.28}$ | $54.79_{\pm 2.01}$ | $25.32_{\pm 0.01}$ |
| 48 | $75.29_{\pm 0.40}$ | $56.28_{\pm 1.46}$ | $38.48_{\pm 2.33}$ |

Table 7: Ablation studies of the repeated sampling iterations $K$ on Adult dataset.

- Weak linear correlation with the target variable: Selected covariates should not exhibit a strong correlation with the target variable. For instance, in the Adult dataset, the attributes marital status, race, and sex were chosen because their Pearson correlation coefficients with the target variable are relatively smaller compared to attributes like *native country* and *workclass*.

- Divergent correlation statistics across training and test datasets: Selected attributes should show notable differences in correlation coefficients between the training and test datasets. For example, in the Taxi dataset, the selected attributes exhibit varying correlation coefficients in the training and test datasets, suggesting that these attributes are not direct causes of the target variable in this context.

## A.6  THE ANALYSIS ON SAMPLE SIZE'S INFLUENCE ON OUR METHOD

Our method use score-based model to model the original data distribution, serving a purpose similar to traditional density estimation methods such as Kernel Density Estimation (KDE). Traditional density estimation methods, such as KDE, often require a large amount of background data. However, thanks to the advantages of diffusion models, our score-based density estimation is not sensitive to sample size, as demonstrated in our subsequent experimental results.

In this section, we examine the impact of training samples' size on our score-based proxy. We create subsets of varying sizes from the original training dataset and use these subsets to train score models. The final classification model is trained on the weights from new score models but tested on the original test data. We denote $R$ as the ratio of the subset size relative to the original dataset. The experimental results are listed in Table 9. It demonstrates that the weights generated from our score-based proxy is insensitive to the sample size. The robustness of our method arises from score model's ability to construct the latent score field. As Kadkhodaie et al. (2024) stated, neural networks can memorize the score field even when the number of training samples is finite. This property ensures that our score-based weights faithfully approximate the original probability density, enabling robust classification model training even with reduced subsets ($R = 0.5$ and $R = 0.8$). However, when $R = 0.2$, the number of training samples becomes insufficient to construct an accurate score field, which makes the estimated weights less accurate. However, even when we used only one-fifth of the data for training, the performance of our model still outperformed the results of the other baselines using the full dataset in Table 1.

In a word, our method could generate more effective weights due to the process of modeling implicit score field, which only requires training a neural network. The process of predicting score for new samples does not require to calculate the interaction with other given training samples like statistical

| Dataset | Attribute | Pearson Coefficient | | Dataset | Attribute | Pearson Coefficient | |
|---|---|---|---|---|---|---|---|
| | | train | test | | | train | test |
| Adult | marital status | -0.0345 | -0.0287 | Taxi | weekday | 0.0200 | -0.0169 |
| | race | -0.0852 | -0.0807 | | month | -0.0004 | -0.0243 |
| | sex | 0.0785 | 0.0700 | | direction | 0.0180 | 0.0754 |
| | *native country* | 0.3872 | 0.3915 | | *distance* | 0.4501 | 0.4618 |
| | *work class* | -0.2160 | -0.2119 | | *hour* | 0.0413 | 0.1220 |
| Default | age | 0.0193 | 0.0224 | Bank | age | -0.0086 | -0.0130 |
| | sex | -0.0396 | -0.0430 | | housing status | -0.0690 | -0.0607 |
| | given credit | -0.0368 | -0.0287 | | marital status | -0.0584 | -0.0757 |
| | *education* | -0.1407 | -0.1365 | | duration | 0.0287 | 0.0263 |
| | *payment* | 0.3297 | 0.2814 | | *job* | 0.2807 | 0.2718 |
| | | | | | *loan* | -0.1395 | -0.1368 |
| Shoppers | traffic type | -0.0548 | -0.0644 | ACS Income (AZ) | race | -0.1127 | -0.1312 |
| | visitor type | -0.0276 | -0.0353 | | sex | -0.1312 | 0.1205 |
| | weekend | 0.0277 | 0.0445 | | *marital status* | 0.2307 | 0.2472 |
| | *administrative* | 0.1422 | 0.1401 | | *age* | 0.2658 | 0.2713 |
| | *administrative duration* | -0.1042 | -0.1030 | | | | |
| ACS Income (MA) | race | -0.1030 | -0.1486 | ACS Income (MI) | race | -0.0620 | -0.0324 |
| | sex | 0.1435 | 0.1086 | | sex | 0.1806 | 0.2030 |
| | *marital status* | 0.2925 | 0.2704 | | *marital status* | 0.2518 | 0.2405 |
| | *age* | 0.2748 | 0.2560 | | *age* | 0.2469 | 0.2127 |

Table 8: The Pearson correlation coefficients between our selected sensitive attributes (with regular font) and the target variable in training and test data. For ease of comparison, we also show some typical attributes (with *italic font*) not selected for evaluation. We set attributes that either (1) exhibit low correlation with the target variable across both training and test datasets, or (2) demonstrate significant variation in correlation with the target variable between the training and test datasets as sensitive (non-causal) attributes. Then we evaluate the model bias concerning these attributes.

density estimation methods. Therefore, the number of training samples has minimal impact on this process, provided the dataset size remains relatively reasonable.

| | $R = 0.2$ | | $R = 0.5$ | | $R = 0.8$ | | $R = 1$, original | |
|---|---|---|---|---|---|---|---|---|
| | Mean | Worst | Mean | Worst | Mean | Worst | Mean | Worst |
| Acc. (%) | $73.81_{\pm1.03}$ | $54.24_{\pm5.48}$ | $74.23_{\pm0.25}$ | $54.57_{\pm1.98}$ | $74.78_{\pm0.62}$ | $54.64_{\pm1.90}$ | $74.33_{\pm0.28}$ | $54.79_{\pm2.01}$ |

Table 9: Experimental performance under different ratio $R$ on Adult dataset. The number of original training and test samples is 32561 and 16281.

### A.7 THE BALANCED SAMPLES AFTER DEPLOYING OUR WEIGHTS

To intuitively demonstrate how our score-based weights balance the original dataset, we divide the samples into groups based on the sensitive attribute and target label, then compute the sum of weights for each group. Without accounting for training challenges caused by group-specific variance, we expect the weights to achieve a balanced distribution. Specifically, the majority group is expected to have a lower weighted sum compared to its unweighted sum. Table 10 presents the results for four groups after applying our score-based weights. For nearly all the listed non-causal attributes, the sum of weights for the majority group decreases, while the sum for the minority group increases. This observation confirms that our weights effectively balance the original distributional shift, offering a clear explanation of how our method operates on real-world datasets.

| sensitive attribute $x$ | Default | | | | | | Shoppers | | | | | |
|---|---|---|---|---|---|---|---|---|---|---|---|---|
| | age | | given credit | | sex | | traffic type | | visitor type | | weekend | |
| | original | weighted | original | weighted | original | weighted | original | weighted | original | weighted | original | weighted |
| $x=1, y=0$ | 9144 | 6943.19 | 9834 | 7390.46 | 12902 | 9766.14 | 4635 | 2298.18 | 8221 | 3951.20 | 2120 | 1025.14 |
| $x=0, y=1$ | 3236 | 5972.06 | 4180 | 7690.34 | 2586 | 4751.89 | 1004 | 3855.73 | 383 | 1413.91 | 1282 | 4930.48 |
| $x=1, y=1$ | 2733 | 5061.28 | 1789 | 3343.00 | 3383 | 6281.44 | 725 | 2707.13 | 1346 | 5148.95 | 447 | 1632.38 |
| $x=0, y=0$ | 11887 | 9023.47 | 11197 | 8576.20 | 8129 | 6200.52 | 4733 | 2235.96 | 1147 | 582.94 | 7248 | 3509 |

Table 10: The number of samples divided into different groups originally as well as the weighted sum of these samples after deploying our score-based weights. The reweighted samples of different groups are more balanced, which is conducive to subsequent unbiased classification learning.

### A.8   THE t-SNE VISUALIZATION COMPARISON WITH OUR SCORE-BASED WEIGHTS

We previously provided an intuition for how our score-based weights help balance datasets using a synthetic example in Figure 1. In this section, we present a straightforward visualization on Default dataset to demonstrate how our method balances the real-world dataset. To be specific, we use t-SNE (Van der Maaten & Hinton, 2008) to reduce the latent representation to two dimensions for visualization. Note that in Table 10, the Default dataset predominantly exhibits label $y$ shift. Therefore, we divide the representations into a majority group ($y = 0$) and a minority group ($y = 1$). In Figure 5a, points from the minority group ($y = 1$) are shaded in dark green, while the majority group ($y = 0$) is colored brown. We could observe that minority samples are often situated in the marginal regions of clusters, indicating that they have low probability densities. Correspondingly, we visualize these samples' score-based weights in Figure 5b. Points with higher weights are represented by warmer colors like purple. We expect that the points from the minority group in Figure 5a to be assigned higher weights in Figure 5b. Comparing these two figures, we can observe that our method significantly increases the weights for samples in the minority group. Almost all the dark green points in the left figure are shaded closer to purple in the right figure, which validates the effectiveness of our score-based weights.

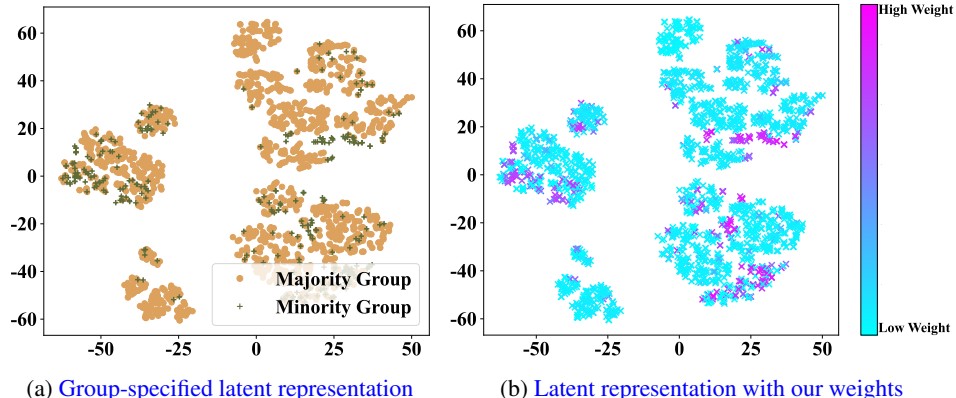

(a) Group-specified latent representation    (b) Latent representation with our weights

Figure 5: The t-SNE visualization of the original latent representations and their corresponding weights in Default dataset.

### A.9   THE COMPARISON WITH STABLE LEARNING

In this section, we provide a comprehensive comparison between our method and stable learning. Stable learning aims to decorrelate features to achieve a uniform and balanced data distribution, which is similar to our approach. However, our method offers three significant advantages over stable learning, as detailed below:

- **Ability to handle potential $Y$-shift problems**: Our method use a similarity difference measure to address implicit $Y$-shift as discussed in Section 3.3. In contrast, stable learning focuses solely on the decorrelation of covariates without differentiating the information carried by labels $y$.

- **Balanced Distribution via Original Distribution Modeling vs. Feature Independence**: Our method obtains a balanced distribution by modeling the original distribution while stable learning achieves balance through feature independence. The reweighting process in stable learning relies on feature decorrelation under a linear assumption. However, it is important to note that **a balanced distribution does not equate to feature independence, and feature decorrelation does not necessarily achieve balance, particularly in non-linear cases.**
  Consider an example where feature decorrelation fails at the sample level but our method succeeds. Suppose there are two features $x_0$ and $x_1$, with samples distributed such that $0 < x_0 < 1$ and $x_0 < x_1 < x_0 + 1$. Suppose that original data distribution is imbalanced, e.g., samples with $x_0 > 0.5$ all share a same higher density than those with $x_0 < 0.5$. Under this circumstance,

our method can transform the original data distribution $p(x_0, x_1)$ into a uniform data distribution $U(x_0, x_1)$ at the **sample level** easily. This is achievable because the score model captures and recovers the original data distribution $p(x_0, x_1)$.

In contrast, stable learning, which enforces feature decorrelation to achieve independence, cannot produce a uniform balanced distribution at the sample level while maintaining independence between $x_0$ and $x_1$. The geometry of the sample space—a parallelogram region—dictates that $x_0$ and $x_1$ cannot be independent while maintaining a uniform sample-level distribution, i.e., $U(x_0, x_1) \neq p(x_0) \cap p(x_1)$. Thus, stable learning fails to balance the data distribution at the sample level while simultaneously decorrelating features.

Decorrelating features to achieve feature independence is a good idea, but it is insufficient to ensure a balanced dataset without the essential assumptions. In contrast, our method does not depend on the presence or absence of feature correlations. Instead, it estimates the implicit score field to model the original distribution, enabling a robust balancing operation based on the estimated distribution.

- **Enhanced Experimental Outcomes Relative to Stable Learning**: To facilitate a quantitative comparison between our method and stable learning, we conducted experiments in Tables 1 and 2. Our approach shows a minimum improvement of 5% in the worst group accuracy compared to the SRDO baseline across both tables. **The superiority of our score-based reweighting over stable learning stems from the accuracy of the weights derived from our score model compared to the predicted probabilities generated by stable learning predictors.** Stable learning methods typically train a predictor to estimate probabilities, which involves the generation of synthetic samples. For instance, SRDO (Shen et al., 2020) employs empty vectors to create samples, whereas StableNet (Zhang et al., 2021) uses a Random Fourier Transformation. The quality of these synthetic samples crucially affects the training process of the predictor, thereby making the estimated probabilities potentially unreliable. In contrast, the score in our method represents the gradient of the log-likelihood of the original distribution $p(x)$. This ensures that our reweighting process remains rigorously faithful to the actual data distribution.

In a word, our method does not rely on any assumptions about the original data distribution. It utilizes the score, i.e., the gradient of estimated log-likelihood of original data distribution, to perform sample reweighting, which is flexible and easy to conduct. The additional experiments further confirm the effectiveness of our method.

## A.10    DISCUSSION ABOUT THE RELATION BETWEEN MEAN AND THE WORST-CASE ACCURACY ON SYNTHETIC DATASET

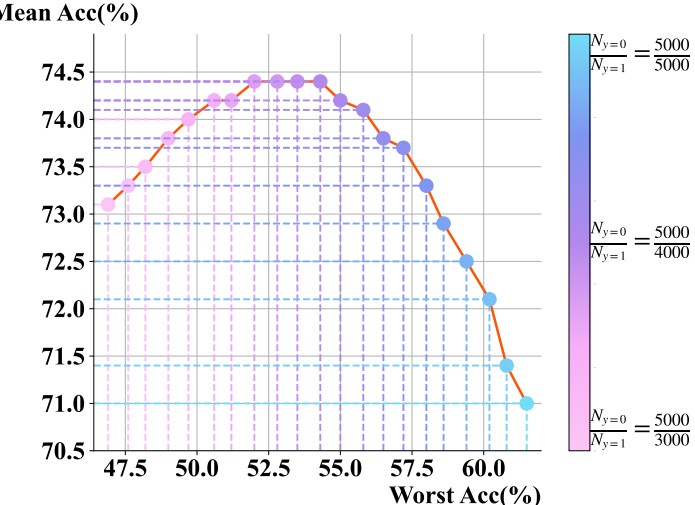

Figure 6: The Pareto curve for the synthetic experiment in Section A.10.

The trade-off between mean accuracy and worst-group accuracy is a well-documented phenomenon. In many cases, a specific optimization objective may prioritize either higher mean accuracy or higher worst-case accuracy. However, we want to emphasize that the trade-off is not always stable, even in synthetic data. To illustrate this point, we conducted a new synthetic experiment inspired by Zhang et al..

We designed a binary classification task with explicit $Y$-shift. Data for each class were generated from two distinct multivariate normal distributions with different means and covariance matrices as listed in Table 11. The sample size for class $y = 0$ was fixed at 5000, while the number of class $y = 1$ samples varied incrementally from 3000 to 5000 in steps of 100. This variation mimicked the effect of reweighting, akin to our score-based balancing approach. Models were trained on these mixtures, each representing a different proportion of $y = 1$ samples. Each experiment was repeated 500 times. The results are visualized in Figure 6.

| Class | Mean | Covariance Matrix | Number of Samples |
|---|---|---|---|
| $y = 0$ | $[-1, 0]$ | $\begin{bmatrix} 5 & 5 \\ 5 & 5 \end{bmatrix}$ | fixed at 5000 |
| $y = 1$ | $[1, 0]$ | $\begin{bmatrix} 15 & 5 \\ 5 & 15 \end{bmatrix}$ | from 3000 to 5000 |

Table 11: Data generation parameters for the synthetic dataset.

In Figure 6, we observe the following:

- **Effect of Dataset Balance on Worst-Case Accuracy:** As the dataset becomes more balanced, the worst-case accuracy increases consistently, mirroring the trend observed with our score-based reweighting strategy. This observation suggests that balancing the dataset is crucial for improving worst-case accuracy.

- **Dynamic Interaction Between Mean and The Worst-Case Accuracy:** When the number of class $y = 1$ samples increases from 3000 to 4000, both mean accuracy and worst-case accuracy increase simultaneously, with no evident trade-off. However, as the number increases from 4000 to 5000, mean accuracy drops significantly. This phenomenon highlights that the trade-off between mean accuracy and worst-case accuracy is not always persistent. Instead, their interaction depends on how the optimization process influences the training trajectories.

### A.11 DISCUSSION ABOUT THE RELATION BETWEEN MEAN AND THE WORST-CASE ACCURACY ON REAL DATASET

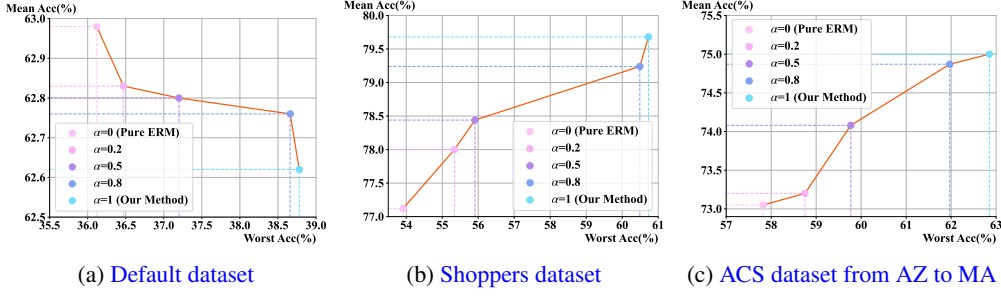

(a) Default dataset     (b) Shoppers dataset     (c) ACS dataset from AZ to MA

Figure 7: The trade-off curve of our method between mean accuracy and worst-group accuracy on three settings which exhibit different kinds of shift.

To better understand how the relationship between average accuracy and worst-group accuracy evolves when optimizing with our score-based weights, we conducted a new experiment and present the Pareto curve in Figure 7.

In fact, by simply combining the loss functions of empirical risk minimization (ERM) and our method, we can achieve a balance between both optimization objectives. Specifically, we defined

a mixed optimization objective as $\mathcal{L}_{\text{mix}} = \alpha \mathcal{L}_{\text{weighted}} + (1 - \alpha)\mathcal{L}_{\text{ERM}}$, where $0 \leq \alpha \leq 1$, and train classification models using $\mathcal{L}_{\text{mix}}$ with varying $\alpha$ values. Here, $\mathcal{L}_{\text{weighted}}$ represents the loss computed with our score-based weights, while $\mathcal{L}_{\text{ERM}}$ corresponds to the standard ERM loss. The parameter $\alpha$ controls the the influence of our weights in the optimization process. As $\alpha$ increases, the optimization objective aligns more closely with our score-based balancing strategy, while a lower $\alpha$ gives greater weight to ERM. By varying $\alpha$, we can compare model performance and gain insights into how these two optimization objectives interact and influence the performance.

We evaluate the trained models under three scenarios: (1) the Default dataset (Column 6-7 in Table 1), (2) the Shoppers dataset (Column 8-9 in Table 1), and (3) training on data from AZ of the ACS income dataset while testing on data from MA (Column 10-11 in Table 2).

Figure 7a reveals a clear trade-off curve between mean accuracy and worst-group accuracy. Notably, the overall trend of the curves forms a near Pareto frontier, supporting the existence of a trade-off between these two accuracies. Furthermore, compared to ERM's standard optimization objective, our method more effectively improves worst-group accuracy.

In Figure 7b, the curve does not form an exact Pareto frontier. Within a certain range, both worst-case accuracy and mean accuracy exhibit similar trends under the distribution shift present in the Shoppers dataset. This suggests that our method can simultaneously enhance both accuracies.

Figure 7c exhibits a curve distinct from Figure 7a. Here, the trade-off between mean and worst-group accuracy is no longer the sole dynamic at play. We attribute this phenomenon to changes in the causal mechanism, specifically $Y|X$-shift caused by selection bias across different states as highlighted in previous studies. From the perspective of optimization, such shifts violate the assumption of independent and identically distributed data, introducing challenges for ERM. Since our method employs a reweighting strategy to balance the dataset, its optimization goal is better suited to this setting than ERM to some extent, resulting in improvements to both mean and worst-group accuracy. However, when compared to methods explicitly designed for scenarios involving causal mechanism changes, our score-based reweighting falls short in achieving the best mean accuracy.

In summary, the relationship between mean accuracy and worst-group accuracy can take many forms. The trade-off between these metrics plays a significant role in optimization, but how this trade-off quantitatively evolves is a complex problem. To the best of our knowledge, there are currently no methods in the community capable of predicting this trend in advance. However, in cases where a trade-off exists, such as in the Default dataset, we can construct a mixed optimization objective combining our loss function and ERM. This allows for control over mean and worst-case accuracy values, as demonstrated in Figure 7a, effectively serving as a "knob" for balancing these metrics. Ultimately, the optimization process determines how mean and worst-case accuracy interact. Notably, a method can outperform another on both metrics if its optimization is better suited to the specific distribution shifts present in the dataset. Our reweighting-based optimization objective is primarily designed to globally optimize for the worst-group accuracy. It consistently achieves the best worst-case accuracy across nearly all evaluated datasets. Additionally, since our method can address both $X$-shift and $Y$-shift through score-based modeling, it performs better than some baselines in terms of mean accuracy under specific types of shifts. These factors collectively contribute to the improved overall performance observed in Column 12-13 of Table 1 and Column 8-9 and 16-17 of Table 2.

