# OpenReview forum: "Latent Score-Based Reweighting for Robust Classification on Imbalanced Tabular Data"
_ICLR.cc/2025/Conference — Submitted to ICLR 2025_

### Official Review · Reviewer_4w6D · 2024-10-28

**Soundness:** 2
**Presentation:** 2
**Contribution:** 2
**Rating:** 6
**Confidence:** 3

**Summary:**

This work introduces a latent score-based reweighting framework to mitigate biases and spurious correlations in machine learning models on tabular data. The method overcomes limitations of existing techniques by capturing the joint data distribution without needing prior group labels and by focusing on balancing the overall data distribution, not just the conditional distribution. It identifies and upweights underrepresented samples based on the similarity of score vectors, thereby enhancing robustness and ensuring a more uniform dataset representation. The approach has been shown to effectively improve performance on imbalanced data across various datasets, even under distribution shifts.

**Strengths:**

1) This paper captures the joint data distribution without the need for prior group labels. It enhances model robustness by ensuring a more uniform dataset representation. Some cases are provided to explain their motivation.
2) The experimental results on various tabular datasets have demonstrated the effectiveness of the proposed method.

**Weaknesses:**

1) Besides boundary-based methods that are mainly compared in this paper, other approaches, like stable learning, are also proposed to balance the overall data distribution. These methods should be also discussed and compared.
2) The design of the method is primarily driven by motivation. The effective generalization bounds require theoretical analysis and proof. In particular, the impact of the error from using proxies instead of density estimates on the model’s generalization capability needs to be explained, along with its limitations.
3) One of the challenges in estimating density estimates is the requirement for a large amount of data; that is, a substantial number of samples are needed to achieve accurate density estimates. It is suggested to investigate whether the proposed proxy of density is independent of the sample size. Relevant experiments, like testing the method on subsets of the datasets with varying sizes and comparing the results,  are recommended to test the effectiveness of the method from this perspective.

**Questions:**

Why is the method proposed in this paper limited to tabular data? Please give the reasons why it is not applicable to other data types, such as images or natural language.

**Details Of Ethics Concerns:**

nan

---

> ### Author Response · Authors · 2024-11-22
> **Response to Reviewer 4w6D (1.1/4)**
>
> We would like to thank you for taking the time to review our paper and for providing insightful comments. Below are responses to the concerns raised by you. Please let us know if you require any further information, or if anything is unclear.
>
> > W1: Besides boundary-based methods that are mainly compared in this paper, other approaches, like stable learning, are also proposed to balance the overall data distribution. These methods should be also discussed and compared.
>
> **[R1]** Thanks for proposing this concern. The idea of stable learning-based method is to make features decorrelated to achieve a balanced data distribution in the sample level, which is similar to our method. However, our method offers three significant advantages over stable learning, as detailed below:
>
> 1. **Ability to handle potential $Y$-shift problems:** Our method use a similarity difference measure to address potential $Y$-shift as discussed in Section 3.3.2. In contrast, stable learning focuses solely on the decorrelation of covariates without differentiating the information carried by labels $y$.
>
> 2. **Balanced Distribution via Original Distribution Modeling vs. Feature Independence:** Our method obtains a balanced distribution by modeling the original distribution while stable learning achieves balance through feature independence. The reweighting process in stable learning relies on feature decorrelation under a linear assumption. However, it is important to note that **a balanced distribution does not equate to feature independence, and feature decorrelation does not necessarily achieve balance, particularly in non-linear cases.**
>
>    Consider an example where feature decorrelation fails at the sample level but our method succeeds. Suppose there are two features $x_0$ and $x_1$, with samples distributed such that $0 < x_0 < 1$ and $x_0 < x_1 < x_0+1$. Suppose that original data distribution is imbalanced, e.g., samples with $x_0 > 0.5$ all share a same higher density than those with $x_0 < 0.5$. Under this circumstance, our method can transform the original data distribution $p(x_0, x_1)$ into a uniform data distribution $U(x_0, x_1)$ at the **sample level** easily. This is achievable because the score model captures and recovers the original data distribution $p(x_0, x_1)$.
>
>    In contrast, stable learning, which enforces feature decorrelation to achieve independence, cannot produce a uniform balanced distribution at the sample level while maintaining independence between $x_0$ and $x_1$. The geometry of the sample space—a parallelogram region—dictates that $x_0$ and $x_1$ cannot be independent while maintaining a uniform sample-level distribution, i.e., $U(x_0, x_1) \neq p(x_0) \cap p(x_1)$. Thus, stable learning fails to balance the data distribution at the sample level while simultaneously decorrelating features.
>
>    Decorrelating features to achieve feature independence is a good idea, but it is insufficient to ensure a balanced dataset without the essential assumptions. In contrast, our method does not depend on the presence or absence of feature correlations. Instead, it estimates the implicit score field to model the original distribution, enabling a robust balancing operation based on the estimated distribution.
>
> 3. **Enhanced Experimental Outcomes Relative to Stable Learning:** To facilitate a quantitative comparison between our method and stable learning, we conducted supplementary experiments and updated the results in Tables 1 and 2 of our revised manuscript. Our approach shows a minimum improvement of 5% in the worst group accuracy compared to the SRDO baseline across both tables. For clarity, SRDO’s performance is highlighted in blue in the revised paper. **The superiority of our score-based reweighting over stable learning stems from the accuracy of the weights derived from our score model compared to the predicted probabilities generated by stable learning predictors.** Stable learning methods typically train a predictor to estimate probabilities, which involves the generation of synthetic samples. For instance, SRDO [1] employs empty vectors to create samples, whereas StableNet [3] uses a Random Fourier Transformation. The quality of these synthetic samples crucially affects the training process of the predictor, thereby making the estimated probabilities potentially unreliable. In contrast, the score in our method represents the gradient of the log-likelihood of the original distribution $p(x)$. This ensures that our reweighting process remains rigorously faithful to the actual data distribution.
>
> In a word, our method does not rely on any assumptions about the original data distribution. It utilizes the score, i.e., the gradient of estimated log-likelihood of original data distribution, to perform sample reweighting, which is flexible and easy to conduct. The additional experiments further confirm the effectiveness of our method.

---

> ### Author Response · Authors · 2024-11-22
> **Response to Reviewer 4w6D (1.2/4)**
>
> Due to word count limitations, we are unable to include the complete table in our "Response to Reviewer 4w6D (1.1/4)". This response provides the experimental results referred in R1.
>
> |      | Adult              |                  | Bank             |                  | Default          |                   | Shoppers         |                  | Taxi             |                  | Average |       |
> | ---- | ------------------ | ---------------- | ---------------- | ---------------- | ---------------- | ----------------- | ---------------- | ---------------- | ---------------- | ---------------- | ------- | ----- |
> |      | Mean               | Worst            | Mean             | Worst            | Mean             | Worst             | Mean             | Worst            | Mean             | Worst            | Mean    | Worst |
> | SRDO | 71.27   $\pm$ 2.43 | 46.44 $\pm$ 6.82 | 66.34 $\pm$ 2.55 | 32.08 $\pm$ 5.44 | 62.38 $\pm$ 8.06 | 33.34 $\pm$ 10.94 | 76.24 $\pm$ 0.35 | 53.11 $\pm$ 2.31 | 64.57 $\pm$ 2.07 | 58.18 $\pm$ 2.50 | 68.16   | 44.63 |
> | Ours | 74.33 $\pm$ 0.28   | 54.79 $\pm$ 2.01 | 69.50 $\pm$ 0.39 | 41.28 $\pm$ 1.56 | 62.62 $\pm$ 0.12 | 38.78 $\pm$ 1.22  | 79.68 $\pm$ 1.44 | 60.73 $\pm$ 3.39 | 67.85 $\pm$ 0.16 | 63.14 $\pm$ 2.02 | 70.80   | 51.74 |
>
>
>
> |      | AZ               |                   | MA               |                  | MI               |                  | Average |       | AZ $\rightarrow$ MA |                  | MA $\rightarrow$ MI |                  | MA $\rightarrow$ MI |                  | Average |       |
> | ---- | ---------------- | ----------------- | ---------------- | ---------------- | ---------------- | ---------------- | ------- | ----- | ------------------- | ---------------- | ------------------- | ---------------- | ------------------- | ---------------- | ------- | ----- |
> |      | Mean             | Worst             | Mean             | Worst            | Mean             | Worst            | Mean    | Worst | Mean                | Worst            | Mean                | Worst            | Mean                | Worst            | Mean    | Worst |
> | SRDO | 75.17 $\pm$ 2.22 | 60.03 $\pm$ 10.23 | 77.85 $\pm$ 1.01 | 69.52 $\pm$ 2.49 | 73.30 $\pm$ 1.22 | 54.63 $\pm$ 5.75 | 75.44   | 61.39 | 74.00 $\pm$ 1.83    | 58.37 $\pm$ 8.57 | 74.78 $\pm$ 0.49    | 61.77 $\pm$ 4.26 | 73.83 $\pm$ 0.35    | 60.05 $\pm$ 0.96 | 74.39   | 60.07 |
> | Ours | 76.28 $\pm$ 0.11 | 66.42 $\pm$ 0.89  | 78.35 $\pm$ 0.14 | 73.33 $\pm$ 1.48 | 75.53 $\pm$ 1.77 | 67.10 $\pm$ 1.61 | 76.72   | 68.95 | 75.00 $\pm$ 0.07    | 62.85 $\pm$ 0.40 | 74.75 $\pm$ 0.35    | 68.75 $\pm$ 0.30 | 74.73 $\pm$ 0.35    | 64.10 $\pm$ 0.35 | 74.83   | 65.23 |
>
>
>
> [1] Shen Z, Cui P, Zhang T, et al. Stable learning via sample reweighting[C]//Proceedings of the AAAI Conference on Artificial Intelligence. 2020, 34(04): 5692-5699.
>
> [2] Kuang K, Cui P, Athey S, et al. Stable prediction across unknown environments[C]//proceedings of the 24th ACM SIGKDD international conference on knowledge discovery & data mining. 2018: 1617-1626.
>
> [3] Zhang X, Cui P, Xu R, et al. Deep stable learning for out-of-distribution generalization[C]//Proceedings of the IEEE/CVF Conference on Computer Vision and Pattern Recognition. 2021: 5372-5382.

---

> ### Author Response · Authors · 2024-11-22
> **Response to Reviewer 4w6D (2/4)**
>
> > W2: In particular, the impact of the error from using proxies instead of density estimates on the model’s generalization capability needs to be explained, along with its limitations.
>
> **[R2]** Thanks for raising this question. We will address this issue step by step.
>
> First, we want to stress that training a score model could be exactly viewed as training a density estimator just like traditional density estimation method. In fact, given a trained score model, the exact probability density could be estimated solely based on score. The detailed explanation is stated in Section 3.3.1. To be specific, score itself, i.e., $\nabla_{\mathbf{x}_t} \text{log}p(\mathbf{x}_t)$ denotes the gradient of the log-likelihood of the original data distribution $p(z)$ at a given noise scale $t$. The diffusion process is modeled as a stochastic differential equation (SDE). As introduced in [1], the corresponding probability flow ordinary differential equation (ODE) is given by: $\mathrm{d}\mathbf{x} = \underbrace{\left[\mathbf{f}(\mathbf{x}, t) - \frac{1}{2}g(t)^2 \mathbf{s} _{\theta}(\mathbf{x}, t) \right]} _{=: \tilde{\mathbf{f}} _\mathbf{\theta}(\mathbf{x}, t)} \mathrm{d}\bar{t}$. By solving this ODE and using the instantaneous change of variables formula, the log-likelihood of the original data distribution could be computed as $\log p _0(\mathbf{x}(0)) = \log p _T(\mathbf{x}(T)) + \int _0^T \nabla \cdot \tilde{\mathbf{f}} _\mathbf{\theta}(\mathbf{x}(t), t) \mathrm{d} t$, **which has the same effect as traditional statistical density estimation method, while leveraging the flexibility and efficiency of score-based modeling.**
>
> Second, we want to clarify that our proxy follows the same trend as the probability likelihood computed via score model's ODE, which is exactly the estimated density. However, **our proxy actually offers numerically more stable sample-level weights compared to the raw likelihood of the probability density**, which makes our approach more suitable for sample-level classification tasks. This is the key question we have discussed in Section 3.3.1 and Section 4.6. Note that our final aim is to balance the datasets globally at the sample level. It requires that the weights of those samples should be within a numerically stable range. While samples can have varying importances, each valid sample must still have an opportunity to contribute to the final classification model. However, the probability density fails to meet this requirement. Extreme differences in log-likelihood values lead to highly imbalanced sample weights, failing to reflect the **relative magnitudes** of densities among different samples globally. Because the log-likelihood strictly follows the functional form of the original data distribution, it leads to explicit numerical differences. When only a few data points have significantly high probability densities, their log-likelihoods become much higher than those of other samples. Using these likelihoods to compute sample weights causes high-density samples to disproportionately overshadow others, whether those have low or moderately high densities. This imbalance diminishes distinctions among samples outside the highest-density regions, reducing the ultimate training on the remaining samples to an unweighted process without meaningful density differentiation.
>
> To validate this point, we conduct the experiment in Section 4.6. In Figure 3a, we could observe that only a small number of points located at cluster centers exhibit significantly higher log-likelihoods. Reweighting based on these estimates could lead to disproportionate emphasis on a small number of extreme high-density points, potentially obscuring the distinction between other high- and low-probability regions. In contrast, our score similarity in Figure 3b alleviates this problem. Our similarity measure not only differentiates density clearly but also maintains a consistent and stable numerical range, making it suitable for sample reweighting.
>
>
> The key difference between our proxy and exact likelihood lies in the interaction between models as defined in Eq. 12. The subtraction operation between outputs of two score models captures the relative difficulty of training models on specific samples. Exact likelihood, on the other hand, reflects only the absolute probability, making it well-suited for generative tasks but less appropriate for sample-level classification tasks. The numerical values of probability density have a scale mismatch with the requirements of sample-level weights for classification, whereas our proxy operates on the same scale as the score model outputs, ensuring consistency and stability in sample reweighting.
>
> [1] Song Y, Sohl-Dickstein J, Kingma D P, et al. Score-based generative modeling through stochastic differential equations[J]. arXiv preprint arXiv:2011.13456, 2020.

---

> ### Author Response · Authors · 2024-11-22
> **Response to Reviewer 4w6D (3/4)**
>
> > W3: One of the challenges in estimating density estimates is the requirement for a large amount of data; that is, a substantial number of samples are needed to achieve accurate density estimates. It is suggested to investigate whether the proposed proxy of density is independent of the sample size. Relevant experiments, like testing the method on subsets of the datasets with varying sizes and comparing the results, are recommended to test the effectiveness of the method from this perspective.
>
>
> **[R3]** Thanks for proposing this concern. Our method use score-based model to model the original data distribution, whose aim is similar to density estimation methods like Kernel Density Estimation (KDE). Traditional density estimation methods, such as KDE, often require a large amount of background data. However, thanks to the advantages of diffusion models, our score-based density estimation is not sensitive to sample size, as demonstrated in our subsequent experimental results.
>
> Here we investigate the influence of training samples' size on our score-based proxy. Subsets of varying sizes were selected from the original training dataset, and score models were trained on these subsets. The final classification model was then trained using the weights generated from these score models but tested on the original test data. We denote $R$ as the ratio of the subset size relative to the original dataset. The experimental results are listed in the following table. It demonstrates that the weights from our score-based proxy are largely insensitive to the sample size. The robustness of our method arises from score model's ability to construct the latent score field.
> As stated in [1], neural networks can memorize the score field even when the number of training samples is finite. This property ensures that our score-based weights faithfully approximate the original probability density, enabling robust classification model training even with reduced subsets ($R=0.5$ or $R=0.8$). However, when $R=0.2$, the number of training samples becomes insufficient to construct an accurate score field, which makes the estimated weights less accurate. However, even when we used only one-fifth of the data for training, the performance of our model still outperformed the results of the other baselines on the full dataset in Table 1 of our paper.
>
> In summary, our method could generate more effective weights due to the process of modeling implicit score field, which only requires training a neural network. The process of predicting score for new samples does not require to calculate the interaction with other given training samples like statistical density estimation methods. Therefore, the number of training samples has minimal impact on this process, provided the dataset size remains relatively reasonable.
>
>
>
> |          | $R=0.2$          |                  | $R=0.5$          |                  | $R=0.8$          |                  | $R=1$ (original) |                  |
> | -------- | ---------------- | ---------------- | ---------------- | ---------------- | ---------------- | ---------------- | ---------------- | ---------------- |
> |          | Mean             | Worst            | Mean             | Worst            | Mean             | Worst            | Mean             | Worst            |
> | Acc. (%) | 73.81 $\pm$ 1.03 | 54.24 $\pm$ 5.48 | 74.23 $\pm$ 0.25 | 54.57 $\pm$ 1.98 | 74.78 $\pm$ 0.62 | 54.64 $\pm$ 1.90 | 74.33 $\pm$ 0.28 | 54.79 $\pm$ 2.01 |
>
>
>
> [1] Kadkhodaie Z, Guth F, Simoncelli E P, et al. Generalization in diffusion models arises from geometry-adaptive harmonic representations[C]//The Twelfth International Conference on Learning Representations.

---

> ### Author Response · Authors · 2024-11-22
> **Response to Reviewer 4w6D (4/4)**
>
> > Q: Why is the method proposed in this paper limited to tabular data? Please give the reasons why it is not applicable to other data types, such as images or natural language.
>
> **[A]** Addressing the issue of distribution shift in tabular data is crucial. Previous studies [1] have highlighted that tabular datasets are often subject to diverse types of distribution shifts. Furthermore, given that each column in tabular data represents specific semantics, ensuring robustness and fairness across sensitive attributes is particularly important. These considerations motivate the development of our method that can maintain robustness under potential covariate and label shifts in tabular datasets.
>
> To address your request, we conducted experiments on the ColorMNIST dataset. Following the original setup described in [2], we excluded the random label-flipping process from [2] to preserve the underlying predictive mechanism. Robustness was evaluated using the worst-group accuracy based on the non-causal feature, i.e., the color of the digits. Each experiment was repeated three times, and we report the mean and standard deviation of the classification accuracies. The results are summarized as follows:
>
> |              | Mean Acc (%)                  | Worst Acc (%)                 |
> | ------------ | ----------------------------- | ----------------------------- |
> | ERM          | $91.20 \scriptsize \pm 3.08$  | $84.85 \scriptsize \pm 9.69$  |
> | CVaR-DRO     | $60.01 \scriptsize \pm 1.54$  | $43.25 \scriptsize \pm 7.70$  |
> | $\chi^2$-DRO | $84.13 \scriptsize \pm 1.72$  | $68.67 \scriptsize \pm 8.49$  |
> | KL-DRO       | $78.84 \scriptsize \pm 13.24$ | $48.10 \scriptsize \pm 14.85$ |
> | EIIL         | $92.37 \scriptsize \pm 2.79$  | $86.95 \scriptsize \pm 5.30$  |
> | JTT          | $92.23 \scriptsize \pm 2.47$  | $85.25 \scriptsize \pm 2.90$  |
> | FAM          | $91.20 \scriptsize \pm 3.08$  | $85.30 \scriptsize \pm 10.32$ |
> | Ours         | $92.73 \scriptsize \pm 3.87$  | $88.00 \scriptsize \pm 10.32$ |
>
> The ColorMNIST dataset includes only one non-causal attribute—namely, the color of the digits—which facilitates validation. Our objective is to improve the worst-group accuracy while maintaining the mean accuracy as much as possible, a goal that our method successfully achieves as demonstrated in the table above. This experiment demonstrates that our approach can be adapted to other types of datasets, provided that the non-causal attributes are identifiable and corresponding validations are feasible. By modeling the original data distribution through score-based methods, our approach enhances robustness without relying on prior information. In summary, our method shows potential for generalization to other data domains. However, this paper focuses on addressing challenges in tabular data, and we leave the exploration of broader generalization to future work.
>
> [1] Liu J, Wang T, Cui P, et al. On the need for a language describing distribution shifts: Illustrations on tabular datasets[J]. Advances in Neural Information Processing Systems, 2024, 36.
>
> [2] Arjovsky M, Bottou L, Gulrajani I, et al. Invariant risk minimization[J]. arXiv preprint arXiv:1907.02893, 2019.

---

> > ### Comment · Reviewer_4w6D · 2024-11-27
> >
> > Thank you for your response. Most of my questions have been addressed, and I plan to raise my score to 6.

---

> > > ### Author Response · Authors · 2024-11-27
> > > **Further Response to Reviewer 4w6D**
> > >
> > > Thank you for your valuable suggestions and constructive feedback. Your insights have significantly contributed to the improvement of our manuscript, and we truly appreciate your support. If you have any further concerns or suggestions, we would be more than happy to engage in additional discussions and make any necessary revisions.

---

### Official Review · Reviewer_f1qS · 2024-10-31

**Soundness:** 1
**Presentation:** 2
**Contribution:** 2
**Rating:** 5
**Confidence:** 5

**Summary:**

The paper studies the distribution shift problem on tabular data. A diffusion model is used to estimate the data density. Then a sample weigthed is computed based on the density score to make the dataset more balanced. The proposed method is novel. Experiments on several tabular datasets with distribution shifts validate the effectivenss of the proposed method.

**Strengths:**

1. The distribution shift problem on tabular data is important.
2. Applying diffusion model to estimate the sample weight to reweight the sample is novel.
3. Experiments results on several classical tabular datasets validate the proposed method could relieve the distribution shift problem.

**Weaknesses:**

1. Motivation is not clear. The title includes the term "tabular data". However, the proposed model does not have any specific design on the tabular data. Moreover, why cannot the model be used in image data? There a lot of image benchmarks can be used to evaluate the problem method, such as WILDS, colored  MNIST, domainbed, waterbirds, celebA, etc.
2. Concept is confused. The manuscript that previous method mainly solves P(Y|X), then the proposed method solve P(X,Y) shift problem (i.e., both P(X) and P(Y|X) problem). To my understanding, the score is estimated only based on P(X), so it could only solve P(X) problem. How the model could handle P(Y|X) problem?
3. Related work has factual error. For the "Achieving robustness with prior information" paragraph, a lots of work do not require the prior information, such as HRM, stable learning, etc. The authors should carefully discuss the difference with them.
4. If the model could only solve the P(X) problem, the stable learning methods could also handle this problem. What is your major advantange?
5. The experimental resutls are weak. The improvement over baselines are marginal. More common used datasets on image should be tested (see W1). More related baselines should be compared, such as methods in [1].
[1] Change is Hard: A Closer Look at Subpopulation Shift

**Questions:**

Please reponse the questions in weaknesses.

---

> ### Author Response · Authors · 2024-11-22
> **Response to Reviewer f1qS (1/5)**
>
> We would like to thank you for taking the time to review our paper and for providing comments. Below are responses to the concerns raised by you. Please let us know if you require any further information, or if anything is unclear.
>
> > Q1: Motivation is not clear. The title includes the term "tabular data". However, the proposed model does not have any specific design on the tabular data. Moreover, why cannot the model be used in image data?
>
> **[A1]** As noted in your comments, addressing the issue of distribution shift in tabular data is crucial. Previous studies [1] have highlighted that tabular datasets are often subject to diverse types of distribution shifts. Furthermore, given that each column in tabular data represents specific semantics, ensuring robustness and fairness across sensitive attributes is particularly important. These considerations motivate the development of our method that can maintain robustness under potential covariate and label shifts in tabular datasets.
>
> To address your request, we conducted experiments on the ColorMNIST dataset. Following the original setup described in [2], we excluded the random label-flipping process from [2] to preserve the underlying predictive mechanism. Robustness was evaluated using the worst-group accuracy based on the non-causal feature, i.e., the color of the digits. Each experiment was repeated three times, and we report the mean and standard deviation of the classification accuracies. The results are summarized as follows:
>
> |              | Mean Acc (%)                  | Worst Acc (%)                 |
> | ------------ | ----------------------------- | ----------------------------- |
> | ERM          | $91.20 \scriptsize \pm 3.08$  | $84.85 \scriptsize \pm 9.69$  |
> | CVaR-DRO     | $60.01 \scriptsize \pm 1.54$  | $43.25 \scriptsize \pm 7.70$  |
> | $\chi^2$-DRO | $84.13 \scriptsize \pm 1.72$  | $68.67 \scriptsize \pm 8.49$  |
> | KL-DRO       | $78.84 \scriptsize \pm 13.24$ | $48.10 \scriptsize \pm 14.85$ |
> | EIIL         | $92.37 \scriptsize \pm 2.79$  | $86.95 \scriptsize \pm 5.30$  |
> | JTT          | $92.23 \scriptsize \pm 2.47$  | $85.25 \scriptsize \pm 2.90$  |
> | FAM          | $91.20 \scriptsize \pm 3.08$  | $85.30 \scriptsize \pm 10.32$ |
> | Ours         | $92.73 \scriptsize \pm 3.87$  | $88.00 \scriptsize \pm 10.32$ |
>
> The ColorMNIST dataset includes only one non-causal attribute—namely, the color of the digits—which facilitates validation. Our objective is to improve the worst-group accuracy while maintaining the mean accuracy as much as possible, a goal that our method successfully achieves as demonstrated in the table above. This experiment demonstrates that our approach can be adapted to other types of datasets, provided that the non-causal attributes are identifiable and corresponding validations are feasible. By modeling the original data distribution through score-based methods, our approach enhances robustness without relying on prior information. In summary, our method shows potential for generalization to other data domains. However, this paper focuses on addressing challenges in tabular data, and we leave the exploration of broader generalization to future work.
>
> [1] Liu J, Wang T, Cui P, et al. On the need for a language describing distribution shifts: Illustrations on tabular datasets[J]. Advances in Neural Information Processing Systems, 2024, 36.
>
> [2] Arjovsky M, Bottou L, Gulrajani I, et al. Invariant risk minimization[J]. arXiv preprint arXiv:1907.02893, 2019.

---

> ### Author Response · Authors · 2024-11-22
> **Response to Reviewer f1qS (2/5)**
>
> > Q2: Concept is confused. The manuscript that previous method mainly solves P(Y|X), then the proposed method solve P(X,Y) shift problem (i.e., both P(X) and P(Y|X) problem). To my understanding, the score is estimated only based on P(X), so it could only solve P(X) problem. How the model could handle P(Y|X) problem?
>
> **[A2]** We want to first clarify the statement in this answer to make our following explanations more clear: In our paper, $P(X)$, $P(Y)$, $P(Y|X)$, and $P(X, Y)$ refer to the data distribution. As for the specific types of distribution shift, we commonly use $X$-shift to indicate covariate shift problem, which arises primarily from  imbalanced data distributions of the covariates $x$. $Y$-shift refers to the imbalances in the outcome labels $y$.
>
> **Returning to your question:** Based on your text in Q4, we guess your question as asking whether our method, since the score model is trained solely on covariates, addresses only the $X$-shift problem. The answer is no. While the $X$-shift problem is the primary focus of our method, it is also capable of addressing the potentia $Y$-shift problem. The reasons are as follows:
>
> - **The types of distribution shift our method tackles:** Our method primarily focuses on addressing $X$-shift at the level of all non-causal covariates, as well as potential $Y$-shift in class labels. The objective is to ensure that the model consistently makes robust predictions across all non-causal covariates rather than optimizing for robustness with respect to a single covariate.
> - **The reason why boundary-based methods may fail:** Our paper claim that previous methods fail to achieve such global robustness because they rely on a pretrained boundary $P(Y|X)$. Here $P(Y|X)$ denotes conditional probability, not the $Y|X$-shift referenced in your text. While under $X$-shift, a balanced training data distribution is required when pursuing a global robust model just like the example in Figure 1. Previous boundary-based methods may fail because they could not model the original training distribution $P(X)$ effectively. In turn, they are unable to balance the training data.
> - **The detailed explanations about how our method utilize the information of $y$ and handle potential $Y$-shift problem:** We guess what you want to ask is that training score model is based on covariates $x$ and how can we model the joint data distribution $P(X, Y)$. Our approach approximates the joint distribution indirectly using the similarity difference measure defined in Eq. 12. As described in Section 3.3.2, we partition the training data by class and train separate score models for each class. The similarity function $\text{Sim}(\cdot)$ in Eq. 11 estimates the probability density within a specific class. For each sample, we compute the similarity difference between score models trained on samples from different classes. The subtraction operation in Eq. 12 ensures that samples from minority classes exhibit lower $\text{SimDiff}$ values compared to those from majority classes, effectively capturing $P(Y)$ indirectly.
>
> In summary, we train score models separately based on the given $y$ labels in the training data and use $\text{SimDiff}$ to approximate $P(X, Y)$. Our similarity difference measure allows us to represent probability density at the sample level. Consequently, our method effectively addresses both $X$-shift at the level of all non-causal covariates and potential $Y$-shift in class labels. If you have any further questions, please do not hesitate to reach out. We are committed to addressing all your concerns.

---

> ### Author Response · Authors · 2024-11-22
> **Response to Reviewer f1qS (3/5)**
>
> > Q3: Related work has factual error. For the "Achieving robustness with prior information" paragraph, a lots of work do not require the prior information, such as HRM, stable learning, etc. The authors should carefully discuss the difference with them.
>
> **[A3]** Thanks for this comment. The "prior information" in this section actually has two distinct meanings:
>
> - **Pure Prior Information**: This refers to explicit prior knowledge that can enhance the training of predictive models, such as **domain labels**.
>
> - **Predefined Prior Assumption**: This represents implicit assumptions about the training data distribution, which underpin the design of the method.
>
> We want to emphasize that while the works you referenced may not explicitly introduce additional covariates, their methodologies are **implicitly** based on various assumptions about the data distribution. This aligns with the second meaning of "prior information" as defined in our text. Detailed explanations are provided below:
>
> - **The optimization process of the second stage in HRM implicitly assumes that samples with the same domain label follow a uniform distribution.** HRM [1,2] implicitly assumes that the samples could be clustered only by data's heteogeneity. Therefore, it generates domain labels to divide subgroups. Since domain labels are discrete, samples with the same domain label are treated equivalently during traning final predictive models. Therefore, the underlying assumption in the final optimization process is that the samples within each cluster conform to **a uniform distribution**.
> - **Stable learning explicitly assumes feature independence could lead to balanced distribution, which is fulfilled in linear situations.** The framework of stable learning methods [2] is inherently built on linear assumptions, targeting specific linear prediction problems. A detailed comparison between stable learning and our method is provided in A4.
>
> In contrast to these methods, our approach neither assumes a specific mathematical form for the original data distribution nor introduces additional covariates. The score-based reweighting mechanism remains entirely faithful to the original given data distribution. We will include these comparisons in Appendix A.9 of our revised paper and provide a detailed explanation of the term "prior information" as requested.
>
> [1] Liu J, Hu Z, Cui P, et al. Heterogeneous risk minimization[C]//International Conference on Machine Learning. PMLR, 2021: 6804-6814.
>
> [2] Shen Z, Cui P, Zhang T, et al. Stable learning via sample reweighting[C]//Proceedings of the AAAI Conference on Artificial Intelligence. 2020, 34(04): 5692-5699.

---

> ### Author Response · Authors · 2024-11-22
> **Response to Reviewer f1qS (4.1/5)**
>
> > Q4: If the model could only solve the P(X) problem, the stable learning methods could also handle this problem. What is your major advantange?
>
> **[A4]** First, we would like to clarify that **our method addresses not only $X$-shift but also potential $Y$-shift**, as detailed in A2 and mentioned in Section 3.3.2 of the main text.
>
> Second, since you referred to stable learning-based methods [1, 2, 3], we provide a comprehensive **empirical** comparison between our method and stable learning along with **analysis experiments**. Stable learning aims to decorrelate features to achieve a uniform and balanced data distribution, which is similar to our approach. However, our method offers three significant advantages over stable learning, as detailed below:
>
> 1. **Ability to handle potential $Y$-shift problems:** Our method use a similarity difference measure to address implicit $Y$-shift as discussed in A2. In contrast, stable learning focuses solely on the decorrelation of covariates without differentiating the information carried by labels $y$.
>
> 2. **Balanced Distribution via Original Distribution Modeling vs. Feature Independence:** Our method obtains a balanced distribution by modeling the original distribution while stable learning achieves balance through feature independence. The reweighting process in stable learning relies on feature decorrelation under a linear assumption. However, it is important to note that **a balanced distribution does not equate to feature independence, and feature decorrelation does not necessarily achieve balance, particularly in non-linear cases.**
>
>    Consider an example where feature decorrelation fails at the sample level but our method succeeds. Suppose there are two features $x_0$ and $x_1$, with samples distributed such that $0 < x_0 < 1$ and $x_0 < x_1 < x_0+1$. Suppose that original data distribution is imbalanced, e.g., samples with $x_0 > 0.5$ all share a same higher density than those with $x_0 < 0.5$. Under this circumstance, our method can transform the original data distribution $p(x_0, x_1)$ into a uniform data distribution $U(x_0, x_1)$ at the **sample level** easily. This is achievable because the score model captures and recovers the original data distribution $p(x_0, x_1)$.
>
>    In contrast, stable learning, which enforces feature decorrelation to achieve independence, cannot produce a uniform balanced distribution at the sample level while maintaining independence between $x_0$ and $x_1$. The geometry of the sample space—a parallelogram region—dictates that $x_0$ and $x_1$ cannot be independent while maintaining a uniform sample-level distribution, i.e., $U(x_0, x_1) \neq p(x_0) \cap p(x_1)$. Thus, stable learning fails to balance the data distribution at the sample level while simultaneously decorrelating features.
>
>    Decorrelating features to achieve feature independence is a good idea, but it is insufficient to ensure a balanced dataset without the essential assumptions. In contrast, our method does not depend on the presence or absence of feature correlations. Instead, it estimates the implicit score field to model the original distribution, enabling a robust balancing operation based on the estimated distribution.
>
> 3. **Enhanced Experimental Outcomes Relative to Stable Learning:** To facilitate a quantitative comparison between our method and stable learning, we conducted supplementary experiments and updated the results in Tables 1 and 2 of our revised manuscript. Our approach shows a minimum improvement of 5% in the worst group accuracy compared to the SRDO baseline across both tables. For clarity, SRDO’s performance is highlighted in blue in the revised paper. **The superiority of our score-based reweighting over stable learning stems from the accuracy of the weights derived from our score model compared to the predicted probabilities generated by stable learning predictors.** Stable learning methods typically train a predictor to estimate probabilities, which involves the generation of synthetic samples. For instance, SRDO [1] employs empty vectors to create samples, whereas StableNet [3] uses a Random Fourier Transformation. The quality of these synthetic samples crucially affects the training process of the predictor, thereby making the estimated probabilities potentially unreliable. In contrast, the score in our method represents the gradient of the log-likelihood of the original distribution $p(x)$. This ensures that our reweighting process remains rigorously faithful to the actual data distribution.
>
> In a word, our method does not rely on any assumptions about the original data distribution. It utilizes the score, i.e., the gradient of estimated log-likelihood of original data distribution, to perform sample reweighting, which is flexible and easy to conduct. The additional experiments further confirm the effectiveness of our method. Please view our revised paper or "Response to Reviewer f1qs (4.2/5)" for experimental results.

---

> ### Author Response · Authors · 2024-11-22
> **Response to Reviewer f1qS (4.2/5)**
>
> Due to word count limitations, we are unable to include the complete table in our "Response to Reviewer f1qS (4.1/5)". This response provides the experimental results referred in A4.
>
> |      | Adult              |                  | Bank             |                  | Default          |                   | Shoppers         |                  | Taxi             |                  | Average |       |
> | ---- | ------------------ | ---------------- | ---------------- | ---------------- | ---------------- | ----------------- | ---------------- | ---------------- | ---------------- | ---------------- | ------- | ----- |
> |      | Mean               | Worst            | Mean             | Worst            | Mean             | Worst             | Mean             | Worst            | Mean             | Worst            | Mean    | Worst |
> | SRDO | 71.27   $\pm$ 2.43 | 46.44 $\pm$ 6.82 | 66.34 $\pm$ 2.55 | 32.08 $\pm$ 5.44 | 62.38 $\pm$ 8.06 | 33.34 $\pm$ 10.94 | 76.24 $\pm$ 0.35 | 53.11 $\pm$ 2.31 | 64.57 $\pm$ 2.07 | 58.18 $\pm$ 2.50 | 68.16   | 44.63 |
> | Ours | 74.33 $\pm$ 0.28   | 54.79 $\pm$ 2.01 | 69.50 $\pm$ 0.39 | 41.28 $\pm$ 1.56 | 62.62 $\pm$ 0.12 | 38.78 $\pm$ 1.22  | 79.68 $\pm$ 1.44 | 60.73 $\pm$ 3.39 | 67.85 $\pm$ 0.16 | 63.14 $\pm$ 2.02 | 70.80   | 51.74 |
>
>
>
> |      | AZ               |                   | MA               |                  | MI               |                  | Average |       | AZ $\rightarrow$ MA |                  | MA $\rightarrow$ MI |                  | MA $\rightarrow$ MI |                  | Average |       |
> | ---- | ---------------- | ----------------- | ---------------- | ---------------- | ---------------- | ---------------- | ------- | ----- | ------------------- | ---------------- | ------------------- | ---------------- | ------------------- | ---------------- | ------- | ----- |
> |      | Mean             | Worst             | Mean             | Worst            | Mean             | Worst            | Mean    | Worst | Mean                | Worst            | Mean                | Worst            | Mean                | Worst            | Mean    | Worst |
> | SRDO | 75.17 $\pm$ 2.22 | 60.03 $\pm$ 10.23 | 77.85 $\pm$ 1.01 | 69.52 $\pm$ 2.49 | 73.30 $\pm$ 1.22 | 54.63 $\pm$ 5.75 | 75.44   | 61.39 | 74.00 $\pm$ 1.83    | 58.37 $\pm$ 8.57 | 74.78 $\pm$ 0.49    | 61.77 $\pm$ 4.26 | 73.83 $\pm$ 0.35    | 60.05 $\pm$ 0.96 | 74.39   | 60.07 |
> | Ours | 76.28 $\pm$ 0.11 | 66.42 $\pm$ 0.89  | 78.35 $\pm$ 0.14 | 73.33 $\pm$ 1.48 | 75.53 $\pm$ 1.77 | 67.10 $\pm$ 1.61 | 76.72   | 68.95 | 75.00 $\pm$ 0.07    | 62.85 $\pm$ 0.40 | 74.75 $\pm$ 0.35    | 68.75 $\pm$ 0.30 | 74.73 $\pm$ 0.35    | 64.10 $\pm$ 0.35 | 74.83   | 65.23 |
>
>
>
> [1] Shen Z, Cui P, Zhang T, et al. Stable learning via sample reweighting[C]//Proceedings of the AAAI Conference on Artificial Intelligence. 2020, 34(04): 5692-5699.
>
> [2] Kuang K, Cui P, Athey S, et al. Stable prediction across unknown environments[C]//proceedings of the 24th ACM SIGKDD international conference on knowledge discovery & data mining. 2018: 1617-1626.
>
> [3] Zhang X, Cui P, Xu R, et al. Deep stable learning for out-of-distribution generalization[C]//Proceedings of the IEEE/CVF Conference on Computer Vision and Pattern Recognition. 2021: 5372-5382.

---

> ### Author Response · Authors · 2024-11-22
> **Response to Reviewer f1qS (5/5)**
>
> > Q5: The experimental results are weak. The improvement over baselines are marginal. More common used datasets on image should be tested (see W1). More related baselines should be compared, such as methods in [1]. [1] Change is Hard: A Closer Look at Subpopulation Shift
>
> **[A5]** For the additional experiments, please refer to A1. For comparisons with related baselines, particularly stable learning, please see A4. We want to emphasize that our primary goal is to enhance overall robustness, as measured by the average of worst-group accuracies, while maintaining the mean accuracy as much as possible. Notably, our method improves the worst-group accuracy by nearly 3% while maintaining minimal impact on the mean accuracy—a remarkable achievement compared to baseline methods.

---

> ### Author Response · Authors · 2024-11-28
> **Looking Forward to Your Further Feedback**
>
> Dear Reviewer f1qS,
>
> Thanks for your valuable efforts in reviewing our paper. Based on your questions, we have revised the paper in the following aspects:
>
> - For W1, we have deployed our method on image data and provided the detailed results in A1.
> - For W2, we have clarified the exact types of distribution shifts our method can address in A2.
> - For W3, we have explained the meaning of "prior information" in our setting and revised Section 2 for better understanding.
> - For W4, we have illustrated our method's advantages over stable learning both analytically and experimentally in Appendix A.9.
> - For W5, we have added an experimental comparison with the proposed baseline and updated Table 1 and 2 accordingly.
>
> We sincerely hope that our previous responses could adequately address your concerns. With the deadline for authors to upload the revised PDF approaching, we would like to kindly inquire if you have further concerns or suggestions. We greatly value this opportunity for dialogue and are always more than happy to engage in further discussions. Provided that your concerns have been well-addressed, we would greatly appreciate it if you could reevaluate our paper and thus consider upgrading your score.
>
> Best regards,
>
> Authors

---

> ### Author Response · Authors · 2024-12-01
> **Sincere Request for a Final Discussion on ICLR Submission 4350**
>
> Dear Reviewer f1qS,
>
> We would like to express our sincere thanks once again for your efforts in reviewing our paper. As there are only two days left for discussion, we would kindly inquire if you have any further suggestions or concerns. We believe we have addressed all the remaining concerns raised by the other reviewers, and we hope that our previous responses have adequatetly addressed your concerns as well. Should you have any further comments or queries, we would greatly appreciate the opportunity to address them promptly.
>
> Best regards,
>
> Authors

---

> > ### Comment · Reviewer_f1qS · 2024-12-02
> > **Thank you for response**
> >
> > Dear authors,
> >
> > Thank you for your detailed response. Some of concerns have been solved. However, I still have the concerns on the following two aspects.
> >
> > 1. What the specific challenges have you solved for "tabular data"?
> > 2. I mean Y|X shift is one of reason to cause P(X,Y) shift. It is better to explicitly describe what kind of shifts you are studied in the text.
> >
> > I would like to raise my score to reflect the changes that authors have made.

---

> > > ### Author Response · Authors · 2024-12-02
> > > **Further Response to Reviewer f1qS**
> > >
> > > We appreciate your review and feedback. We are pleased to know that we have addressed most of your concerns. Our responses to the remaining points are as follows:
> > >
> > > > Rebuttal-Q1: What the specific challenges have you solved for "tabular data"?
> > >
> > > **[Rebuttal-A1]** We would like to clarify that our decision to use tabular data is not driven by unique challenges associated with this data type. Tabular data is universally recognized as important, and it is particularly prone to distribution shifts, making the need to ensure robustness under such shifts a significant and widely acknowledged challenge. Given these critical issues, applying our approach to tabular data and presenting results exclusively on tabular datasets is sufficient to warrant a stand-alone paper.
> > >
> > > Our novel score-based approach is designed to maintain robustness under both covariate shifts ($X$-shift) and label shifts ($Y$-shift) without requiring prior information. In addition to demonstrating superiority over baseline methods on the tabular data we focused on, our method is also applicable to other types of data. As shown in the experiments on a CV dataset that we conducted in our previous response A1, our method indeed has the potential to achieve robustness across various data forms. We plan to explore broader generalization in future work. However, if you require experiments on additional datasets, we would be happy to include them in the final version of our paper.
> > >
> > > > Rebuttal-Q2: I mean Y|X shift is one of reason to cause P(X,Y) shift. It is better to explicitly describe what kind of shifts you are studied in the text.
> > >
> > > **[Rebuttal-A2]** Thank you for raising this question. We would like to clarify that the shifts we aim to address are covariate shift ($X$-shift) and potential label shift ($Y$-shift), as stated in Sections 3.3.2 and 4.1 of our original paper. While $Y|X$-shift is not the central concern of our manuscript, we have also conducted experiments to demonstrate how our methods behave both under $Y|X$-shift and non-$Y|X$-shift conditions. The results are discussed in Section 4.5.2, with a more detailed analysis provided in the newly added section Appendix A.11.
> > >
> > > As the deadline for submitting revisions has passed, we are currently unable to update the manuscript. However, we assure you that we will revise the paper once it is received, to clarify more explicitly the types of distribution shifts our method can handle, beyond the current version.
> > >
> > > ------
> > >
> > > We would like to express our sincere gratitude for your effort in reviewing our paper. We believe that these discussions will significantly enhance the coherence of our paper, ensuring readers without background knowledge understand our idea. Provided that your concerns have been well-addressed based on all our comments, we would greatly appreciate it if you could consider upgrading your score to a positive one. Should you have any further suggestions, please feel free to reach out. We are more than happy to engage in further discussions.

---

> > > > ### Comment · Reviewer_f1qS · 2024-12-03
> > > >
> > > > Thank you for the further clarification. You acknowledge that you do not make any specific design on tabular data. Hence, it is inapproprate to have the word "tabular data" in the title.
> > > >
> > > > Moreover, I have concerns on why a uniform training distribution might offer better generalization performance. A uniform distribution can cause the dataset to lose its original information. Further justification is needed regarding the correctness of towarding a "uniform distribution". This is a more critical question.

---

> > > > > ### Author Response · Authors · 2024-12-04
> > > > > **Final Response to Reviewer f1qS (1/3)**
> > > > >
> > > > > Thanks for your concerns. Please see our responses as follows:
> > > > >
> > > > > > Rebuttal-Q2-1:  You acknowledge that you do not make any specific design on tabular data. Hence, it is inappropriate to have the word "tabular data" in the title.
> > > > >
> > > > > **[Rebuttal-R2-1]**
> > > > >
> > > > > We would like to clarify that our use of "tabular data" is intended to **avoid potential overclaim**, rather than to draw attention. Given the significance of distribution shift problems in tabular data, we deployed our method on such data to verify its effectiveness (though we have also confirmed its applicability to a CV dataset in our previous response A1). Therefore, we chose to emphasize "tabular data" in our title to clearly reflect the scope of our paper. In summary, our experiments conducted on tabular data alone justify a stand-alone paper as stated in our last response Rebuttal-A1. **We will revise our paper according to the feedbacks from reviewers to ensure the description is clear.**

---

> > > > > ### Author Response · Authors · 2024-12-04
> > > > > **Final Response to Reviewer f1qS (3/3)**
> > > > >
> > > > > > The reason why we use score-based weights to achieve sample-level uniform distribution.
> > > > >
> > > > > **[Rebuttal-R2-2-2]** Please note that the main objective of this paper is to ensure the robustness along all sensitive attributes. It is the **worst group accuracy** that directly reflects this goal. Commonly, the test dataset is composed of multiple subpopulations, and we only focus on the worst-performing subpopulation. Given that we have no prior information about the test dataset, the most fundamental way to ensure worst-group accuracy is maintaining fairness among the potential training subgroups. This requires balance and "uniformity" among the training samples. This idea aligns with the widely-acknowledged importance sampling, where samples are weighted according to their rarity or importance to achieve a more balanced representation [1].
> > > > >
> > > > > Here we want to give an example to illustrate how a balanced dataset promotes representing the worst group. Suppose there are two subgroups in a dataset, learning one subgroup will not influence the learning for another and the difficulty for learning each sample is the same. We assume that the risk of the model’s performance on each subgroup decreases as the total weights of the samples in that group used during training increase. If the total weights of these two groups are balanced, then the risk of each group will also be the same, making each of them the worst group. In this case, we record the worst group risk of this balanced dataset as $r _u$. However, once there are differences between these two groups, the risk of minority group $r _{\text{min}}$ will become larger than the majority group $r _{\text{max}}$, which makes the worst group risk equal to $r _{\text{min}}$. Obviously, $r _{\text{min}}$ is also larger than $r_u$ since its total weights are less than the weights from balanced groups. Our reweighting process could also be explained by this idea. Ideally, the best weights assigned to samples could be expressed as $w(X) = \frac{P _{\text{test}}(X)}{P _{\text{train}}(X)}$. However, clearly we don't know $P _{\text{test}}(X)$. In this case, if we want to enhance the worst group accuracy, the fundamental way is making sure $P _{\text{train}}(X)$ of each potential subgroup equal, which requires a balanced dataset.
> > > > >
> > > > > To achieve the goal of obtaining a balanced dataset, we propose using our score-based method to adjust the weights of training dataset. This approach ensures that all potential subpopulations are treated near equally during the training of predictive models, indirectly enhancing robustness in the worst group.
> > > > >
> > > > > [1] Chawla N V, Bowyer K W, Hall L O, et al. SMOTE: synthetic minority over-sampling technique[J]. Journal of artificial intelligence research, 2002, 16: 321-357.
> > > > >
> > > > > > The problem that "a uniform distribution can cause the dataset to lose its original information".
> > > > >
> > > > > **[Rebuttal-R2-2-3]** Considering that you referred to stable learning-based methods in your original comment, we guess you are suggesting that **a uniform marginal distribution for each feature, achieved through feature independence**, can cause the dataset to lose its original **significant** information. This statement is correct. In fact, all of the reweighting methods (including the baseline work stable learning recommended by you) will cause the loss of original information of a dataset more or less. However, please note the following points:
> > > > >
> > > > > - **Losing information is not always detrimental to the objective. Sometimes, reweighting may help removing biased information, which in turn promotes the robustness.** Since our aim is to improve robustness as referred in Rebuttal-R2-2-2, the key lies in dropping the imbalanced and biased information to ensure worst-group performance. To achieve this, we ensure **sample-level balance** by estimating the distribution using score models. In this context, removing biased information actually contributes to ensuring robustness.
> > > > > - **Unlike stable learning-based methods, our approach does not remove all the correlations between features.** Methods that perform feature decorrelation (e.g., stable learning) or attempt to enforce uniform marginal distributions of each feature may lose the original information **too much**. In fact, this is exactly one of the advantages our method surpass stable learning-based methods, as we highlighted in our previous response A4 and Appendix A.9. To illustrate this point more straightforwardly, please recall the example in Rebuttal-R2-2-1-3.
> > > > >
> > > > >
> > > > > In summary, our method does not attempt to make every feature follow a uniform distribution. Instead, we use score-based weights to achieve a sample-level balanced training dataset, ensuring that all potential subpopulations are treated similarly. This approach enhances robustness across all sensitive attributes in the test dataset, as validated through our experiments. We hope these explanations address your concerns.

---

> ### Author Response · Authors · 2024-12-02
> **Sincere Request for Your Final Feedback**
>
> Dear Reviewer f1qS,
>
> We sincerely appreciate your efforts in reviewing our paper. As the Author-Reviewer discussion period is drawing to a close, we would like to kindly inquire if you have any additional suggestions. We are more than willing to engage in further discussions to address any potential concerns.
>
> Best regards,
>
> Authors

---

> ### Author Response · Authors · 2024-12-04
> **Final Response to Reviewer f1qS (2/3)**
>
> > Rebuttal-Q2-2: Moreover, I have concerns on why a uniform training distribution might offer better generalization performance. A uniform distribution can cause the dataset to lose its original information. Further justification is needed regarding the correctness of towarding a "uniform distribution". This is a more critical question.
>
> **[Rebuttal-R2-2]** Thanks for raising this concern. We would like to address it point by point.
>
> > The meaning of "uniform" in our paper.
>
> **[Rebuttal-R2-2-1]**
>
> - **The "uniform" in our paper denotes sample-level balance.**
>
> **[Rebuttal-R2-2-1-1]** We want to first clarify that the "uniform" in our paper refers to the **sample level**, which aims to achieve a balanced joint distribution $P(X)$. Please note that this is different from the "uniform" at the **feature level**, which requires every feature to fulfill a uniform marginal distribution $U(x_i)$. We want to emphasize that these two concepts are fundamentally different. Sample-level uniformity is critical for ensuring robustness, and we will explain the rationale in Rebuttal-R2-2-2. In contrast, a uniform distribution at the feature level would indeed cause the dataset to lose its original significant information for generalization and we will discuss it in Rebuttal-R2-2-3.
>
> - **A balanced, sample-level-uniformly-distributed dataset is not the same as a dataset where all features follow a uniform marginal distribution.**
>
> **[Rebuttal-R2-2-1-2]** We would like to point out that **a reweighted dataset whose joint distribution is balanced does not necessarily imply that all its features will follow a uniform distribution separately**. Moreover, uniform-distributed marginal distribution does not always indicate a balanced joint distribution. Here we will provide an example, please refer to this [link](https://anonymous.4open.science/r/5E3B/uniform%20explanation.pdf) for a detailed illustration (try downloading the PDF if the webpage can't exhibit correctly). Here we have a dataset which contains six samples in the left figure, and four of them are totally the same. If we aim for a balanced dataset at the sample level, we can just enhance the weights for those two minority sample as four times large as majority samples just like the situation in the right figure. Note that in this case, the dataset becomes balanced through reweighting, while $p(x_0)$ obviously does not fulfill a uniform distribution.
>
> - **Achieving a balanced, sample-level-uniformly-distributed dataset does not require feature independence, and in fact, feature independence may not promote a balanced dataset.**
>
> **[Rebuttal-R2-2-1-3]** We want to clarify that deploying reweighting operation to achieve a balanced dataset does not require feature independence. In addition, some operations, such as decorrelating features and promoting feature independence just like stable learning-based methods you have referred in Q4, could not assure a balanced dataset. Here we would like to detail the example where feature decorrelation fails at the sample level but our method succeeds. The corresponding illustration is in this [link](https://anonymous.4open.science/r/5E3B/advantages%20over%20stable%20learning.pdf). Suppose there are two features $x _0$ and $x _1$, with samples distributed such that $0<x _0<1$ and $x _0<x _1<x _0+1$. Suppose that original data is imbalanced, e.g., samples with $x _0>0.5$ all share a same higher density than those with $x _0<0.5$. Under this circumstance, our method can transform the original data distribution $p(x _0,x _1)$ into a uniform joint distribution $U(x _0,x _1)$ at the **sample level** easily. This is achievable because the score model captures and recovers the original data distribution $p(x _0,x _1)$. In contrast, the geometry of the sample space—a parallelogram region—dictates that $x _0$ and $x _1$ cannot be independent while maintaining a uniform sample-level distribution, i.e., $U(x _0,x _1) \neq p(x _0)\cap p(x _1)$. In addition, if $x _0$ and $x _1$ both fulfills a uniform distribution, the joint distribution $p(x _0,x _1)$ will not be balanced, which also confirms our Rebuttal-R2-2-1-2.
>
> In summary, our "uniformity" refers to the balanced joint distribution, which is different from the feature-level "uniformity". Therefore, the drawbacks brought by feature-level uniformity do not match with our method, which will be detailed in Rebuttal-R2-2-3. Understanding the distinction between the two meanings of "uniform" in this context is critical. We will include a discussion of the different interpretations of "uniform" in the final version of our paper.

---

### Official Review · Reviewer_fTYk · 2024-11-03

**Soundness:** 3
**Presentation:** 3
**Contribution:** 3
**Rating:** 8
**Confidence:** 3

**Summary:**

The paper attempts to improve robustness by modeling the joint distribution and taking a global perspective on re-weighing as opposed to the popular methods which mostly focus on rectifying the samples near the decision boundary. The observation of most current robustness methods are limited by the pre-trained classification boundaries $P(Y \mid X)$ adds a fresh perspective to the robustness literature. The paper has solid experiments and ablations to support its claims and it is well written.

**Strengths:**

- [S1] Well motivated paper, clearly highlighting the issue in current literature that the methodology is attempting to mitigate.
- [S2] The demo code seems well written and simple enough to run and adapt.
- [S3] I really like the idea proposed by the authors as it intuitively seems to overcome quite a few problems, the experiments and ablations provide good support to the approach.

**Weaknesses:**

- [W1] While I like the motivation presented in terms of the sinusoidal function and the multi-modal gaussian, it would be interesting to look at a more real world scenario where one can expect the proposed method to suit the setup better than traditional methods. That would further help strengthen the case.
- [W2] Table 1 does not include variance values in the main paper, it would be helpful to contrast the effectiveness of the method across various runs, given the general discussion about variance in other parts of the paper as well. Effectively combining Table 5 and Table 6 in the Table 1 and Table 2 or referencing the variation with some reformatting could be possible in the given space. Even moving Algorithm 1, potentially at the cost of Related Work might be a good idea.
- [W3] While the visualization in Figure 3 helps gain insight into how the method works as a proxy, it would be interesting to see some analytical reasoning behind why this might work at scale for larger datasets.

**Questions:**

- [Q1] I’m unsure of how, despite being 2nd best on the mean performance in 2 columns in Table 2 (Columns 11 to 19) the authors claim the method is not best in this scenario due to variations in the predictive mechanism as opposed to Table 1, where the method is not the best (or even top 2) on 3 datasets. So the story doesn’t translate as well, do the authors have justifications other than the intuition as to why they made this claim? It is not obvious how the change in the causal mechanism is a potential cause for variation in the predictive mechanism; more details on this would help clarify the statement.
- [Q2] I am also slightly curious about the formalism behind Section A.4, is there prior literature investigating or stating these are the non-causal attributes? If so are all other attributes not selected necessary causal? The selection of these attributes plays a big role in the methodology, so including correlation statistics or defining a proper heuristic for selection of these attributes would help make a stronger case.
- [Q3] Is there an intuitive way to extend this approach to multiclass classification? Would it be computationally feasible?

---

> ### Author Response · Authors · 2024-11-22
> **Response to Reviewer fTYk (1/4)**
>
> We would like to thank you for providing detailed and helpful comments. In addition, we are pleased that you like the idea of this paper. Please see our response to your concerns:
> > W1: While I like the motivation presented in terms of the sinusoidal function and the multi-modal gaussian, it would be interesting to look at a more real world scenario where one can expect the proposed method to suit the setup better than traditional methods. That would further help strengthen the case.
>
> **[R1]** Thanks for proposing this question. Our similarity measure is designed to generate sample-level weights which could help balance the datasets with distribution shift. To intuitively show that our score-based weights effectively promote balance and facilitate robust model training, we have added a new section A.7 in our revised paper. The following content provides the detailed explanation.
>
> To verify the effectiveness of our score-based weights, we take a straightforward approach: dividing samples by their sensitive attributes and target labels, then computing the sum of weights for each group. Ignoring the challenges of training models due to group variance, we expect the weights to achieve balance. Specifically, the majority group’s weighted sum is expected to decrease relative to its unweighted sum, while the minority group’s weighted sum should increase.
>
> The results for four groups after applying our score-based weights are presented in the following table. In almost all cases, the weighted sum for the majority group decreases, while the minority group’s weighted sum increases. This confirms that our weights effectively balance the original distribution shift, offering a clear explanation of how our method performs on real-world datasets. These findings suggest that our approach is suitable for addressing imbalances in non-synthetic datasets, and the score-based similarity measure can reliably balance datasets at the sample level. We are willing to provide more experimental results if you have any other concern.
>
> |                         | Default  |          |              |          |          |          | Shoppers     |          |              |          |          |          |
> | ----------------------- | -------- | -------- | ------------ | -------- | -------- | -------- | ------------ | -------- | ------------ | -------- | -------- | -------- |
> | Sensitive Attribute $x$ | age      |          | given credit |          | sex      |          | traffic type |          | visitor type |          | weekend  |          |
> |                         | original | weighted | original     | weighted | original | weighted | original     | weighted | original     | weighted | original | weighted |
> | $x$=1, $y$=0            | 9144     | 6943.19  | 9834         | 7390.46  | 12902    | 9766.14  | 4635         | 2298.18  | 8221         | 3951.20  | 2120     | 1025.14  |
> | $x$=0, $y$=1            | 3236     | 5972.06  | 4180         | 7690.34  | 2586     | 4751.89  | 1004         | 3855.73  | 383          | 1413.91  | 1282     | 4930.48  |
> | $x$=1, $y$=1            | 2733     | 5061.28  | 1789         | 3343.00  | 3383     | 6281.44  | 725          | 2707.13  | 1346         | 5148.95  | 447      | 1632.38  |
> | $x$=0, $y$=0            | 11887    | 9023.47  | 11197        | 8576.20  | 8129     | 6200.52  | 4733         | 2235.96  | 1147         | 582.94   | 7248     | 3509     |
>
> > W2: Table 1 does not include variance values in the main paper, it would be helpful to contrast the effectiveness of the method across various runs, given the general discussion about variance in other parts of the paper as well. Effectively combining Table 5 and Table 6 in the Table 1 and Table 2 or referencing the variation with some reformatting could be possible in the given space. Even moving Algorithm 1, potentially at the cost of Related Work might be a good idea.
>
> **[R2]** Thank you for your comments. As you correctly noted, we have reported all standard deviations for our experiments in the initial version of the paper. However, due to page constraints and concerns about visual clarity, we initially presented these tables separately. Combining them into a single table at that time would have resulted in a cluttered layout and excessively small font sizes. However, we have carefully considered your feedback, and revisions are currently underway. We are confident that our next version will address your concerns and fulfill your requirements.

---

> ### Author Response · Authors · 2024-11-22
> **Response to Reviewer fTYk (2/4)**
>
> > W3: While the visualization in Figure 3 helps gain insight into how the method works as a proxy, it would be interesting to see some analytical reasoning behind why this might work at scale for larger datasets.
>
> **[R3]** This is an excellent question. As requested, we have added t-SNE visualizations in a new Section A.8. Please refer to Figure 5 in our revised paper. Specifically, we provide a straightforward visualization on the Default dataset to demonstrate that our method effectively balances real-world datasets.
>
> In the last table in R1, the Default dataset primarily exhibits label ($y$) shift. Thus, we divide the representations into a majority group ($y=0$) and a minority group ($y=1$). In Figure 5.a, points from the minority group ($y=1$) are shaded in dark green, while the majority group ($y=0$) is colored brown. We observe that minority samples are often situated in the marginal regions of clusters, indicating that they have low probability densities. Correspondingly, we visualize the score-based weights of these samples in Figure 5.b, with points assigned higher weights represented by warmer colors (e.g., purple).
>
> We expect the minority group points in Figure 5.a to be assigned higher weights in Figure 5.b. Comparing the two figures, we observe that our method significantly increases the weights for samples in the minority group. Almost all dark green points in Figure 5.a are shaded closer to purple in Figure 5.b, validating the effectiveness of our score-based weighting approach.
>
>
> > Q1: I’m unsure of how, despite being 2nd best on the mean performance in 2 columns in Table 2 (Columns 11 to 19) the authors claim the method is not best in this scenario due to variations in the predictive mechanism as opposed to Table 1, where the method is not the best (or even top 2) on 3 datasets. So the story doesn’t translate as well, do the authors have justifications other than the intuition as to why they made this claim? It is not obvious how the change in the causal mechanism is a potential cause for variation in the predictive mechanism; more details on this would help clarify the statement.
>
> **[A1]** Thanks for proposing this question and your observation is very insightful! As discussed in Section 4.2, our primary evaluation metric is the worst-group accuracy, as our goal is to achieve robust predictions across all non-causal covariates. It is widely acknowledged in the research community that there is often a trade-off between worst-group accuracy and mean accuracy. Including mean accuracy in our tables serves to demonstrate that our method does not achieve higher worst-group accuracy at the cost of a substantial drop in mean accuracy. As shown, our method achieves the best worst-group accuracy across nearly all datasets, though it does not always achieve the highest mean accuracy. This explains why, in Table 1, some mean accuracy results for our method are not the best.
>
> To address your question regarding the mean accuracy results in Table 2, we specifically highlight the ACS dataset because it demonstrates strong selection bias, as identified in the WhyShift study [1]. This study revealed that the ACS datasets exhibit differing causal mechanisms based on spatial location, meaning the predictive relationship between features and labels is not consistent across states. In this case, the instability in mean accuracy is influenced not only by the trade-off between mean and worst-group accuracy but also by shifts in causal relationships. Therefore, we report this factor in our paper to distinguish the results in columns 11–19 of Table 2 from those in Table 1 and columns 2–10 of Table 2.
>
> [1] Liu J, Wang T, Cui P, et al. On the need for a language describing distribution shifts: Illustrations on tabular datasets[J]. Advances in Neural Information Processing Systems, 2024, 36.

---

> ### Author Response · Authors · 2024-11-22
> **Response to Reviewer fTYk (3/4)**
>
> > Q2: I am also slightly curious about the formalism behind Section A.4, is there prior literature investigating or stating these are the non-causal attributes? If so are all other attributes not selected necessary causal? The selection of these attributes plays a big role in the methodology, so including correlation statistics or defining a proper heuristic for selection of these attributes would help make a stronger case.
>
> **[A2]** This is an excellent concern. As you mentioned, selecting non-causal attributes is crucial for evaluating robustness globally. To address this, we present the Pearson correlation coefficients between the attributes and the target variable in both the training and test datasets in the following table.
>
> | Dataset             | Attribute                 | Train   | Test    | Dataset             | Attribute        | Train   | Test    |
> | ------------------- | ------------------------- | ------- | ------- | ------------------- | ---------------- | ------- | ------- |
> | **Adult**           | marital status            | -0.0345 | -0.0287 | **Taxi**            | weekday          | 0.0200  | -0.0169 |
> | **Adult**           | race                      | -0.0852 | -0.0807 | **Taxi**            | month            | -0.0004 | -0.0243 |
> | **Adult**           | sex                       | 0.0785  | 0.0700  | **Taxi**            | direction        | 0.0180  | 0.0754  |
> | **Adult**           | *native country*          | 0.3872  | 0.3915  | **Taxi**            | *distance*       | 0.4501  | 0.4618  |
> | **Adult**           | *work class*              | -0.2160 | -0.2119 | **Taxi**            | *hour*           | 0.0413  | 0.1220  |
> | **Default**         | age                       | 0.0193  | 0.0224  | **Bank**            | age              | -0.0086 | -0.0130 |
> | **Default**         | sex                       | -0.0396 | -0.0430 | **Bank**            | housing status   | -0.0690 | -0.0607 |
> | **Default**         | given credit              | -0.0368 | -0.0287 | **Bank**            | marital status   | -0.0584 | -0.0757 |
> | **Default**         | *education*               | -0.1407 | -0.1365 | **Bank**            | duration         | 0.0287  | 0.0263  |
> | **Default**         | *payment*                 | 0.3297  | 0.2814  | **Bank**            | *job*            | 0.2807  | 0.2718  |
> | **Shoppers**        | traffic type              | -0.0548 | -0.0644 | **Bank**            | *loan*           | -0.1395 | -0.1368 |
> | **Shoppers**        | visitor type              | -0.0276 | -0.0353 | **ACS Income (AZ)** | race             | -0.1127 | -0.1312 |
> | **Shoppers**        | weekend                   | 0.0277  | 0.0445  | **ACS Income (AZ)** | sex              | -0.1312 | 0.1205  |
> | **Shoppers**        | *administrative*          | 0.1422  | 0.1401  | **ACS Income (AZ)** | *marital status* | 0.2307  | 0.2472  |
> | **Shoppers**        | *administrative duration* | -0.1042 | -0.1030 | **ACS Income (AZ)** | *age*            | 0.2658  | 0.2713  |
> | **ACS Income (MA)** | race                      | -0.1030 | -0.1486 | **ACS Income (MI)** | race             | -0.0620 | -0.0324 |
> | **ACS Income (MA)** | sex                       | 0.1435  | 0.1086  | **ACS Income (MI)** | sex              | 0.1806  | 0.2030  |
> | **ACS Income (MA)** | *marital status*          | 0.2925  | 0.2704  | **ACS Income (MI)** | *marital status* | 0.2518  | 0.2405  |
> | **ACS Income (MA)** | *age*                     | 0.2748  | 0.2560  | **ACS Income (MI)** | *age*            | 0.2469  | 0.2127  |
>
> The Pearson correlation coefficients between our selected sensitive attributes (shown in regular font) and the target variable in both training and test datasets are presented in the following table. For comparison, we also include some typical attributes (shown in *italic font*) that were not selected for evaluation. Sensitive attributes are chosen for evaluation if they meet at least one of the following conditions:
>
> 1. **Weak Linear Correlation with the Target Variable:** The selected covariates exhibit weak correlations with the target variable. For instance, in the Adult dataset, our selected attributes—marital status, race, and sex—show relatively smaller Pearson coefficients compared to attributes such as *native country* and *workclass*.
> 2. **Differences in Correlation Statistics Between Training and Test Datasets:** Selected attributes exhibit varying correlation coefficients across training and test datasets, suggesting that these attributes are not direct causes of the target variable. For example, in the Taxi dataset, the selected attributes demonstrate significant differences in correlation coefficients between training and test datasets, reinforcing their selection as non-causal in this context.
>
> These details have been added to Appendix A.5 of our revised paper.

---

> ### Author Response · Authors · 2024-11-22
> **Response to Reviewer fTYk (4/4)**
>
> > Q3: Is there an intuitive way to extend this approach to multiclass classification? Would it be computationally feasible?
>
> **[A3]** Thanks for raising the concern. For multiclass classification problems, suppose there are $C$ classes denoted by $Y=\{y_0, y_1, ..., y_{C-1}\}$, we could still use $\text{SimDiff}$ to differentiate densities. Similar to the binary classification scenario, we train $C$ separate score models, one for each class. Then for a data point $z_i$ with label $y_i$, $\text{Sim}(z_i;F_{y=y_i})$ is computed using the score model $F_{y=y_i}$ corresponding to its class.
>
> Next, $\text{Sim}$ is calculated for the other $C-1$ score models, and the mean of these $C-1$ values is used to represent  $\text{Sim}(z_i;F_{y \neq y_i})$. As a result, Eq.12 is modified as follows:
>
> $\text{SimDiff}(z_i) = \frac{ \sum_{y' \in Y \setminus \{y_{z_i}\}}  \text{Sim}(z_i; F_{y = y'})}{C-1} - \text{Sim}(z_i; F_{y = y_{z_i}})$.
>
> For binary classification problems ($C=2$), this equation naturally reduces to the original Eq. 12 in the main text.
>
> It is important to note that training a score model for a specific label $y_m$ requires only the samples belonging to that class. Therefore, the total computational cost for training the score models scales linearly with the total number of training samples, ensuring no additional computational overhead.
>
> We will update these details in the revised manuscript and leave further discussions for future works.

---

> > ### Comment · Reviewer_fTYk · 2024-11-22
> > **Reply to the Rebuttal**
> >
> > I appreciate the thorough rebuttal provided by the authors and enumerate my thoughts below.
> >
> > - [R1] The reweighing numbers provided in the table for Default and Shopper do corroborate the authors' claim and I like the new appendix section A.7
> > - [R2] Noted.
> > - [R3] Figure 5 looks quite promising! Appendix 8 does allay my concerns about the direct applicability to larger real-world scale datasets.
> > - [A1] While I understand the goal of the authors' is to ensure best worst-case accuracy, I am not sure how the objective would actually lead to that? I further agree that ensuring a superior worst-case accuracy without losing out on mean accuracy is important, but is there a knob of sorts? Is there a possibility of seeing a Pareto frontier for the worst-case v/s average accuracy and how over-optimizing for one can negatively affect the other? Further, while I understand where the non-best mean average performance in Column 11 to 19 comes from in Table 2, but the method achieves the best worst-case nonetheless. The justification for this phenomenon, while seems correct intuitively, I presume there is no way of explicitly disentangling the causes of the drop in mean accuracy right? Would there be a particular tradeoff point where mean accuracy would be higher, but worst-case maybe worse? A Pareto curve might help clear these concerns.
> > - [A2] Great! This adds a lot of weight to the story, appendix section A.5 looks perfect to justify the choice now.
> > - [A3] That seems reasonable. Anything beyond is out of the scope of the paper anyways in my opinion.
> >
> > I've gone through the other reviews and rebuttals, I personally believe reporting results just on tabular datasets is sufficient as a stand-alone paper in itself.
> >
> > But I do look forward to the discussion period and will keep my score and lean acceptance for now :)

---

> > > ### Author Response · Authors · 2024-11-24
> > > **Further Response to Reviewer fTYk (1/3)**
> > >
> > > We appreciate your detailed review and insightful feedback. We are pleased to know that we have addressed most of your concerns. Our responses to the remaining points (Q1&A1) are as follows:
> > >
> > > As you correctly stated, the trade-off between mean accuracy and worst-group accuracy is a well-documented phenomenon. In many cases, a specific optimization objective may prioritize either higher mean accuracy or higher worst-case accuracy. However, we want to emphasize that the trade-off is not always stable, even in synthetic data. To illustrate this point, we conducted a new synthetic experiment inspired by [1, 2].
> > >
> > > We designed a binary classification task with explicit $Y$-shift. Data for each class were generated from two distinct multivariate normal distributions with different means and covariance matrices. The sample size for class $y=0$ was fixed at 5000, while the number of class $y=1$ samples varied incrementally from 3000 to 5000 in steps of 100. This variation mimicked the effect of reweighting, akin to our score-based balancing approach. Models were trained on these mixtures, each representing a different proportion of $y=1$ samples. Each experiment was repeated 500 times, and the mean and worst-case accuracies of the trained models on test data were reported. The detailed experimental setup can be found in Appendix A.10 of the revised paper. The results are summarized in the table below and visualized in Figure 6 of the revised paper.
> > >
> > > | $N_{y=0}$ | $N_{y=1}$ | Mean Accuracy     | Worst Accuracy    |
> > > | --------- | --------- | ----------------- | ----------------- |
> > > | 5000      | 3000      | 0.731 $\pm$ 0.009 | 0.469 $\pm$ 0.017 |
> > > | 5000      | 3100      | 0.733 $\pm$ 0.010 | 0.476 $\pm$ 0.018 |
> > > | 5000      | 3200      | 0.735 $\pm$ 0.009 | 0.482 $\pm$ 0.017 |
> > > | 5000      | 3300      | 0.738 $\pm$ 0.009 | 0.490 $\pm$ 0.017 |
> > > | 5000      | 3400      | 0.740 $\pm$ 0.009 | 0.497 $\pm$ 0.017 |
> > > | 5000      | 3500      | 0.742 $\pm$ 0.009 | 0.506 $\pm$ 0.016 |
> > > | 5000      | 3600      | 0.742 $\pm$ 0.009 | 0.512 $\pm$ 0.016 |
> > > | 5000      | 3700      | 0.744 $\pm$ 0.009 | 0.520 $\pm$ 0.016 |
> > > | 5000      | 3800      | 0.744 $\pm$ 0.009 | 0.528 $\pm$ 0.015 |
> > > | 5000      | 3900      | 0.744 $\pm$ 0.010 | 0.535 $\pm$ 0.015 |
> > > | 5000      | 4000      | 0.744 $\pm$ 0.010 | 0.543 $\pm$ 0.014 |
> > > | 5000      | 4100      | 0.742 $\pm$ 0.010 | 0.550 $\pm$ 0.014 |
> > > | 5000      | 4200      | 0.741 $\pm$ 0.011 | 0.558 $\pm$ 0.015 |
> > > | 5000      | 4300      | 0.738 $\pm$ 0.011 | 0.565 $\pm$ 0.015 |
> > > | 5000      | 4400      | 0.737 $\pm$ 0.011 | 0.572 $\pm$ 0.015 |
> > > | 5000      | 4500      | 0.733 $\pm$ 0.012 | 0.580 $\pm$ 0.015 |
> > > | 5000      | 4600      | 0.729 $\pm$ 0.012 | 0.586 $\pm$ 0.014 |
> > > | 5000      | 4700      | 0.725 $\pm$ 0.012 | 0.594 $\pm$ 0.014 |
> > > | 5000      | 4800      | 0.721 $\pm$ 0.011 | 0.602 $\pm$ 0.013 |
> > > | 5000      | 4900      | 0.714 $\pm$ 0.012 | 0.608 $\pm$ 0.014 |
> > > | 5000      | 5000      | 0.710 $\pm$ 0.012 | 0.615 $\pm$ 0.013 |
> > >
> > > In Figure 6 and the table above, we observe the following:
> > >
> > > - **Effect of Dataset Balance on Worst-Case Accuracy:** As the dataset becomes more balanced, the worst-case accuracy increases consistently, mirroring the trend observed with our score-based reweighting strategy. This observation suggests that balancing the dataset is crucial for improving worst-case accuracy.
> > >
> > > - **Dynamic Interaction Between Mean and Worst-Case Accuracy:** When the number of class $y=1$ samples increases from 3000 to 4000, both mean accuracy and worst-case accuracy increase simultaneously, with no evident trade-off. However, as the number increases from 4000 to 5000, mean accuracy drops significantly. This phenomenon highlights that the trade-off between mean accuracy and worst-case accuracy is not always persistent. Instead, their interaction depends on how the optimization process influences the training trajectories.
> > >
> > > With the above understanding, in our previous response, we attributed the inability of our method to achieve the best mean accuracy on certain datasets, such as Default in Table 1, to the trade-off between mean accuracy and worst-case accuracy. Back to your new question: is there a possibility of seeing a Pareto frontier for the worst-case v/s average accuracy? To explore this further, we conducted a new experiment to analyze how the relationship between average accuracy and worst-group accuracy evolves when optimizing with our score-based weights. The results are presented as a Pareto curve in Figure 7 of our revised paper and the detailed description of the experiment is listed as follows.

---

> > > ### Author Response · Authors · 2024-11-24
> > > **Further Response to Reviewer fTYk (2/3)**
> > >
> > > In fact, by simply combining the loss functions of empirical risk minimization (ERM) and our method, we can achieve a balance between both optimization objectives. Specifically, we defined a mixed optimization objective as $\mathcal{L} _{\text{mix}} = \alpha \mathcal{L} _{\text{weighted}} + (1 - \alpha)\mathcal{L} _{\text{ERM}}$, where $0 \leq \alpha \leq 1$, and train classification models using $\mathcal{L} _{\text{mix}}$ with varying $\alpha$ values. Here, $\mathcal{L} _{\text{weighted}}$ represents the loss computed with our score-based weights, while $\mathcal{L} _{\text{ERM}}$ corresponds to the standard ERM loss. The parameter $\alpha$ controls the the influence of our weights in the optimization process. As $\alpha$ increases, the optimization objective aligns more closely with our score-based balancing strategy, while a lower $\alpha$ gives greater weight to ERM. By varying $\alpha$, we can compare model performance and gain insights into how these two optimization objectives interact and influence the performance.
> > >
> > > We evaluate the trained models under three scenarios: (1) the Default dataset (Column 6\~7 in Table 1), (2) the Shoppers dataset (Column 8\~9 in Table 1), and (3) training on data from AZ of the ACS income dataset while testing on data from MA (Column 10\~11 in Table 2).
> > >
> > > Figure 7a reveals a clear trade-off curve between mean accuracy and worst-group accuracy. Notably, the overall trend of the curves forms a near Pareto frontier, supporting the existence of a trade-off between these two accuracies. Furthermore, compared to ERM's standard optimization objective, our method more effectively improves worst-group accuracy.
> > >
> > > In Figure 7b, the curve does not form an exact Pareto frontier. Within a certain range, both worst-case accuracy and mean accuracy exhibit similar trends under the distribution shift present in the Shoppers dataset. This suggests that our method can simultaneously enhance both accuracies.
> > >
> > > Figure 7c exhibits a curve distinct from Figure 7a. Here, the trade-off between mean and worst-group accuracy is no longer the sole dynamic at play. We attribute this phenomenon to changes in the causal mechanism, specifically $Y|X$-shift caused by selection bias across different states as highlighted in previous studies. From the perspective of optimization, such shifts violate the assumption of independent and identically distributed data, introducing challenges for ERM. Since our method employs a reweighting strategy to balance the dataset, its optimization goal is better suited to this setting than ERM to some extent, resulting in improvements to both mean and worst-group accuracy. However, when compared to methods explicitly designed for scenarios involving causal mechanism changes, our score-based reweighting falls short in achieving the best mean accuracy.
> > >
> > > In summary, the relationship between mean accuracy and worst-group accuracy can take many forms. The trade-off between these metrics plays a significant role in optimization, but how this trade-off quantitatively evolves is a complex problem. To the best of our knowledge, there are currently no methods in the community capable of predicting this trend in advance. However, in cases where a trade-off exists, such as in the Default dataset, we can construct a mixed optimization objective combining our loss function and ERM. This allows for control over mean and worst-case accuracy values, as demonstrated in Figure 7a, effectively serving as a "knob" for balancing these metrics. Ultimately, the optimization process determines how mean and worst-case accuracy interact. Notably, a method can outperform another on both metrics if its optimization is better suited to the specific distribution shifts present in the dataset. Our reweighting-based optimization objective is primarily designed to globally optimize for the worst-group accuracy, as discussed in our previous responses R1 and R3. It consistently achieves the best worst-case accuracy across nearly all evaluated datasets. Additionally, since our method can address both $X$-shift and $Y$-shift through score-based modeling, it performs better than some baselines in terms of mean accuracy under specific types of shifts. These factors collectively contribute to the improved overall performance observed in Column 12\~13 of Table 1 and Column 8\~9 and 16\~17 of Table 2.

---

> > > ### Author Response · Authors · 2024-11-24
> > > **Further Response to Reviewer fTYk (3/3)**
> > >
> > > We would like to thank you once again for raising these concerns. We believe that these discussions significantly enhance the rigor and soundness of our paper. Please feel free to reach out with any additional questions or points for discussion—we are more than happy to engage further.
> > >
> > > [1] Zhang F, He Q, Kuang K, et al. Distributionally Generative Augmentation for Fair Facial Attribute Classification[C]//Proceedings of the IEEE/CVF Conference on Computer Vision and Pattern Recognition. 2024: 22797-22808.
> > >
> > > [2] Sagawa S, Raghunathan A, Koh P W, et al. An investigation of why overparameterization exacerbates spurious correlations[C]//International Conference on Machine Learning. PMLR, 2020: 8346-8356.

---

> ### Comment · Reviewer_fTYk · 2024-12-02
>
> I appreciate the efforts the authors have put in for clarifying my doubt, the mixed optimization objective is indeed interesting. While this could be a whole research direction in itself, the interplay between mean and worst case accuracy is an interesting discussion and going beyond what the authors presented seems a bit out of scope.
>
> I'm happy to recommend acceptance for the paper :)
>
> P.S. Have updated the rating to accept accordingly.

---

> > ### Author Response · Authors · 2024-12-02
> > **Sincere Thanks to Reviewer fTYk**
> >
> > We would like to express our sincere gratitude for your valuable suggestions and supportive feedback. Your previous suggestions have significantly contributed to enhancing the quality of our manuscript. We appreciate it much! We are also pleased to have already addressed all of your concerns. It was a pleasure discussing the details of our paper with you during the rebuttal. Should you have any additional suggestions, we are more than willing to engage in further discussions and make any necessary improvements to the paper.

---

### Official Review · Reviewer_BAGe · 2024-11-04

**Soundness:** 3
**Presentation:** 4
**Contribution:** 4
**Rating:** 6
**Confidence:** 4

**Summary:**

This paper proposes a latent score-based reweighting framework for improving robustness of classification accuracy under distribution shift. Score-based model (diffusion model) on latent representations of data distribution is used to estimate the joint data distribution P(X,Y), so that low density regions are upweighted to ensure a more balanced representation during training. The proposed method differs from previous methods since it does not require prior information and it models the joint data distribution without making assumptions on the pre-trained classification boundary of P(Y|X). Experiments on various tabular datasets under distribution shift demonstrate that the proposed method improves worse case classification accuracy as compared to baseline methods.

**Strengths:**

This paper provides a novel method that solves an important task and works well in practice by performing experiments across synthetic data and several real datasets.

**Weaknesses:**

1. Diffusion model is used to model latent covariates X. SimDiff addresses y shift by taking the difference between between two similarity metrics across two different classes. However, the paper claims that it directly estimates the joint data distribution P(X,Y). It would be better to make the connection clearer.
2. Average similarity measure is proposed based on an intuition derived from a simulation example with mixtures of two Gaussians. Is the intution generalizable to more general settings? If the intuition is not satisfied in all settings, how would it affect the performance of the method?

**Questions:**

See weakness section above.

---

> ### Author Response · Authors · 2024-11-22
> **Response to Reviewer BAGe**
>
> We would like to thank you for providing helpful comments and positive feedbacks. Following are point to point response to address your concern.
> > Q1: Diffusion model is used to model latent covariates X. SimDiff addresses y shift by taking the difference between between two similarity metrics across two different classes. However, the paper claims that it directly estimates the joint data distribution P(X,Y). It would be better to make the connection clearer.
>
> **[A1]** Thanks for proposing this concern. We would like to clarify that our method does not directly model $P(X, Y)$. Instead, we approximate it with $\text{SimDiff}$, as referenced in your question. $\text{SimDiff}$ integrates the information from $y$ by differentiating samples with different labels and training separate score models for each $y$. As stated in Section 3.3, we approximate $P(X)$ within each class with the similarity measure $\text{Sim}$. The subtraction operation in Eq. 12 ensures that samples from minority classes exhibit lower $\text{SimDiff}$ values compared to those from majority classes, indirectly capturing $P(Y)$. Therefore, the label information $y$ is incorporated by computing the similarity difference, $\text{SimDiff}$, across different class labels. In summary, while $\text{SimDiff}$ serves as a proxy for the joint distribution $P(X, Y)$, it effectively captures and reflects the relative relationships among labels $y$. As a result, it provides sufficient information to generate sensible weights which help balance the imbalanced dataset.
>
> > Q2: Average similarity measure is proposed based on an intuition derived from a simulation example with mixtures of two Gaussians. Is the intution generalizable to more general settings? If the intuition is not satisfied in all settings, how would it affect the performance of the method?
>
> **[A2]** Thanks for proposing this question. Our similarity measure is designed to generate sample-level weights which could help balance the datasets with distribution shift. To intuitively show that our score-based weights effectively promote balance and facilitate robust model training, we have added a new section A.7 in our revised paper. The following content provides the detailed explanation.
>
> To verify the effectiveness of our score-based weights, we take a straightforward approach: dividing samples by their sensitive attributes and target labels, then computing the sum of weights for each group. Ignoring the challenges of training models due to group variance, we expect the weights to achieve balance. Specifically, the majority group’s weighted sum is expected to decrease relative to its unweighted sum, while the minority group’s weighted sum should increase.
>
> The results for four groups after applying our score-based weights are presented in the following table. In almost all cases, the weighted sum for the majority group decreases, while the minority group’s weighted sum increases. This confirms that our weights effectively balance the original distribution shift, offering a clear explanation of how our method performs on real-world datasets. These findings suggest that our approach is suitable for addressing imbalances in non-synthetic datasets, and the score-based similarity measure can reliably balance datasets at the sample level. We are willing to provide more experimental results if you have any other concern.
>
> |                         | Default  |          |              |          |          |          | Shoppers     |          |              |          |          |          |
> | ----------------------- | -------- | -------- | ------------ | -------- | -------- | -------- | ------------ | -------- | ------------ | -------- | -------- | -------- |
> | Sensitive Attribute $x$ | age      |          | given credit |          | sex      |          | traffic type |          | visitor type |          | weekend  |          |
> |                         | original | weighted | original     | weighted | original | weighted | original     | weighted | original     | weighted | original | weighted |
> | $x$=1, $y$=0            | 9144     | 6943.19  | 9834         | 7390.46  | 12902    | 9766.14  | 4635         | 2298.18  | 8221         | 3951.20  | 2120     | 1025.14  |
> | $x$=0, $y$=1            | 3236     | 5972.06  | 4180         | 7690.34  | 2586     | 4751.89  | 1004         | 3855.73  | 383          | 1413.91  | 1282     | 4930.48  |
> | $x$=1, $y$=1            | 2733     | 5061.28  | 1789         | 3343.00  | 3383     | 6281.44  | 725          | 2707.13  | 1346         | 5148.95  | 447      | 1632.38  |
> | $x$=0, $y$=0            | 11887    | 9023.47  | 11197        | 8576.20  | 8129     | 6200.52  | 4733         | 2235.96  | 1147         | 582.94   | 7248     | 3509     |

---

### Author Response · Authors · 2024-11-22
**General Responses to All Reviewers**

We sincerely thank all reviewers for their thoughtful feedback and for recognizing our paper as well-written (Reviewers BAGe, fTYk), driven by a novel motivation (Reviewers BAGe, fTYk, f1qS), and validated effectively on real-world datasets (All Reviewers). We are also particularly pleased that our examples and illustrations helped most reviewers grasp the key ideas of our work.

We acknowledge all reviewers' questions. The main requests and concerns can be summarized as follows:

1. Provide more specific details on how the experimental validation was conducted.
2. Investigate whether our score-based method is sensitive to sample size, as traditional density estimation methods often are.
3. Explain how our intuition generalizes to real-world cases.
4. Compare our method with stable learning, another typical reweighting-based approach.
5. Adapt our method to other types of data formats.

We have carefully addressed these concerns in our revised paper. The major revisions are as follows:

- **For (1):** We have added essential details on how sensitive attributes were selected, as outlined in Appendix A.5.
- **For (2):** We conducted new experiments and reviewed past studies, as presented in Appendix A.6. These results demonstrate that our method is inherently robust to sample size variations, a property stemming from score-based modeling.
- **For (3):** We provided both quantitative analyses (Appendix A.7) and visual illustrations (Appendix A.8) to explain how our method generalizes to real-world scenarios.
- **For (4):** We highlighted three main advantages of our approach over stable learning-based methods in Appendix A.9. Additionally, we applied a typical stable learning method SRDO to all datasets and updated the corresponding experimental results in the revised paper.
- **For (5):** We conducted additional experiments on an image dataset and shared the results with corresponding reviewers. Our method outperformed all baselines in these tests. However, it is important to emphasize that the focus of this paper is on tabular data, which represents a significant and practical domain of application. Thus, we will reserve the discussion of results on other data types for future work.

To facilitate review, we have color-coded all revisions in the updated manuscript. If any reviewer has additional concerns or suggestions, please feel free to reach out—we are always willing to engage in further discussion.

Best regards,

The Authors

---

### Author Response · Authors · 2024-11-26
**Request for Further Discussions**

We sincerely appreciate the insightful questions and helpful suggestions from all the reviewers. We believe our responses have addressed all the concerns raised so far, but we are more than happy to provide any further clarifications if needed. As the discussion period has already passed the halfway point, we kindly invite reviewers to share any additional questions or suggestions. We deeply value this opportunity for dialogue and are eager to engage further to ensure all concerns are thoroughly addressed.

Best regards,

The Authors

---

### Meta-Review · Area_Chair_91Pn · 2024-12-14

**Metareview:**

This paper describes a new approach to handle distribution shift by reweighing the framework. The proposed approach is motivated by the need to mitigate biases and spurious correlations in machine learning models on tabular data. The method identifies and upweights underrepresented samples based on the similarity of score vectors to ensure an uniform dataset representation. The approach has been shown to effectively improve robustness over tabular datasets.

### Recommendation

Reviewers engaged with this submission and identified strengths in its topic and approach. Following the rebuttal and discussion, the overall sentiment among reviewers was "weakly" positive. Given the borderline evaluation, I also reviewed the submission as well -- considering the original submission, the supplementary materials, the reviews, and the responses.

Overall, my recommendation is to reject the paper at this time. My recommendation is primarily based on a number of weaknesses that affect clarity, significance, and potentially soundness. Some of these issues could be addressed through a round of revisions but should be validated through an additional round of peer review.

### Weakness 1: Soundness/Significance/Limitations

The proposed approach is primarily validated through experiments on real-world data where the authors examine accuracy. The paper demonstrates gains. However, it does not sufficiently explore the underlying drivers or the required conditions. This is concerning in several respects.

1. Significance - i.e., readers who wish to adopt this method may not be convinced, especially given that other approaches rest on firm theoretical foundations

2. Limitations - i.e., practitioners may adopt the proposed method without understanding the potential risk. This seems concerning as the effectiveness arises under causal assumptions which can be misspecified.

3. Validity - i.e., in the worst case, we may be observing an artifact. This is a minor point but one that is entirely possible given that we are examining summary statistics on a handful of experiments.

One way to address this would be to perform a synthetic study where we compare the method to alternatives on a simple 2D problem where we observe ground truth. This study can build intuition and highlight potential limitations. Stepping back, reweighing is practical but has several limitations (e.g., potential for misspecification,  and impact on other metrics).

### Weakness 2: Claims/Clarity

The discussion highlights several key points of confusion among reviewers. Many of these issues pertain to the central claims of the paper – e.g., which kinds of distribution shifts does this handle, why would other methods not work, why is this limited to tabular datasets? Some of these issues were resolved through discussion, but should be integrated into an updated paper. Overall, the claims should be measured and supported through evidence.

**Additional Comments On Reviewer Discussion:**

See above.

---

### Decision · Program_Chairs · 2025-01-22

Reject